# Differentially Private Synthetic Data Generation with Diversity via APIs

## Abstract

Synthetic data has emerged as a key solution for preserving the privacy of original data in fields dealing with sensitive information, such as healthcare and finance. Recent advancements in foundation models have significantly improved the quality of synthetic data. However, most high-performance foundation models are only available as black-box APIs, limiting fine-tuning capabilities and requiring private data containing sensitive information to be transmitted to external servers. To address this issue, PE was introduced as a privacy-preserving synthetic data generation method that leverages genetic algorithms with black-box foundation models. Nevertheless, due to its evolutionary process, PE tends to repeatedly focus on a limited subset of samples, leading to a significant reduction in the diversity of the generated synthetic dataset. Since diversity is a crucial factor for enhancing the utility of synthetic data and ensuring robustness across various scenarios, we propose Div-PE, an improved approach that overcomes the diversity limitations of PE through a sample-variant two-stage voting mechanism. This method enhances data diversity and yields a 17.2% gain in FID and an 11.0% increase in downstream accuracy on ResNet-18, averaged over ImageNet, Camelyon17, and UTKFace. Furthermore, Div-PE demonstrates its versatility by delivering strong experimental results not only on image data but also across other modalities, including tabular and text data, validating its applicability to a wide range of data types.

## 1 Introduction

With the rapid improvement of AI performance and the broadening of its applications, concerns over privacy violations through AI have also increased (Lee et al., 2024; Achuthan et al., 2024; Zhan et al., 2025). In domains such as finance and healthcare, where sensitive information is frequently handled, regulatory and legal restrictions often render training datasets for AI models not publicly available. One promising solution that has drawn significant attention is the use of synthetic data. (Assefa et al., 2020; Schreyer et al., 2019). Synthetic data are generated to mimic only the statistical distribution of real data without directly containing personal information (Lu et al., 2023), and have been successfully used as an alternative to private data to address privacy concerns (Mendes et al., 2025; Jordon et al., 2022; Gonzales et al., 2023; Arora et al., 2025; Qian et al., 2024; Nisevic et al., 2025; Kaabachi et al., 2025; Balch et al., 2024; Potluru et al., 2023). However, even though synthetic data do not explicitly include individual records and are therefore safer than real data, rare cases or unique distributional features can still be exposed, leaving a risk of re-identification (Haim et al., 2022; Fredrikson et al., 2015; Choquette-Choo et al., 2021; Tramèr et al., 2022; Wang et al., 2023). To overcome these limitations, Differential Privacy (DP) (Dwork, 2006) has been widely applied. Early approaches added noise to SGD gradients (Abadi et al., 2016), and more recent methods apply DP to large foundation models (Lin et al., 2023; Xie et al., 2024) to generate synthetic data while mathematically guaranteeing that the inclusion of any individual cannot be inferred.

Applying DP to the synthetic data generation process introduces several constraints. In particular, recent studies have confirmed the presence of scaling laws across generative AI (Henighan et al., 2020; Aghajanyan et al., 2023; Fan et al., 2024; Kaplan et al., 2020; Rosenfeld, 2021). Within this context, foundation models trained on massive datasets with enormous computational resources exhibit strong general-purpose performance across diverse domains (OpenAI, 2023; Rombach et al., 2022; Betker et al., 2023; Touvron et al., 2023; Dubey et al., 2024; Anthropic, 2023; DeepMind,

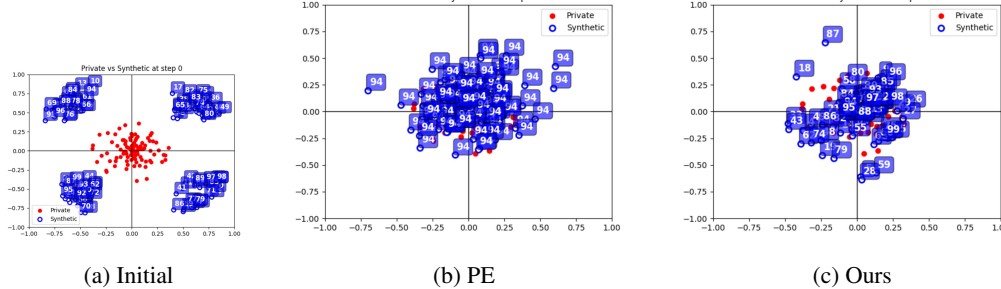

(a) Initial  (b) PE  (c) Ours

Figure 1: Performance comparison in a toy example with PCA visualization. We assigned unique ancestry markers to samples in the initial population (a), and made them pass on their marker to descendants when generating the next generation. At $T = 17$, Div-PE (c) ensures that all ancestral lineages survive, while PE (b) sees one lineage (originating from ancestor 94) dominate all descendants.

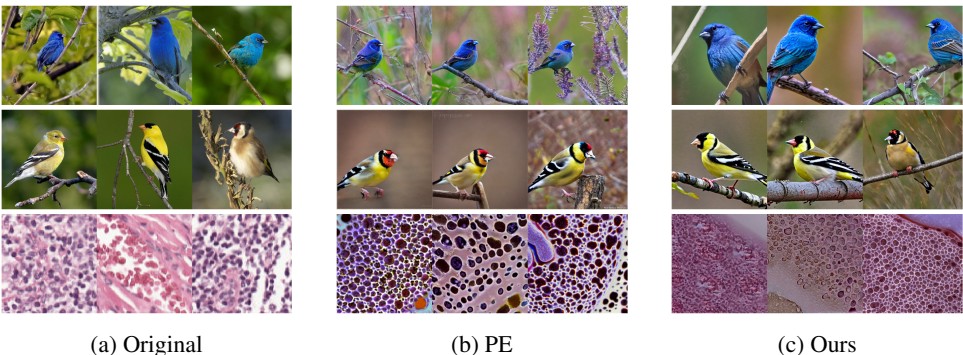

(a) Original  (b) PE  (c) Ours

Figure 2: Original and generated images $(8.24, 10^{-2}$-DP) from ImageNet (top 2 rows) and Camelyon17 (bottom).

2023; Services, 2023; OpenAI, 2025), making it increasingly difficult for task-specific models to keep pace. Consequently, both research and industrial applications are converging toward the use of foundation models (Qin et al., 2024; Awais et al., 2025; Yuan, 2023). The challenge is that many state-of-the-art (SOTA) foundation models are provided only as black-box APIs (OpenAI, 2023; Anthropic, 2023; DeepMind, 2023; Services, 2023; OpenAI, 2025). This prevents fine-tuning, limiting domain adaptability and complicating the application of DP methods that rely on gradient descent, such as DPSGD (Abadi et al., 2016). Alternatives such as prompt engineering (Chen et al., 2023) and prefix tuning (Li & Liang, 2021) provide partial customization, , yet transmitting training data to external API servers introduces privacy risks such as eavesdropping and hijacking (He et al., 2023; Li et al., 2022; Ghalebikesabi et al., 2023; Yue et al., 2023; Harder et al., 2023; 2021; Tang et al., 2024).

To overcome these limitations, Private Evolution (PE) (Lin et al., 2023) was introduced as a framework for privacy-preserving synthetic data generation in black-box settings, drawing inspiration from evolutionary algorithms (Holland, 1975). The process begins by generating an initial population and creating $K$ variations of each sample. The fitness of each sample is then evaluated by averaging the similarities of its variations, weighted by how many private samples select it as their nearest neighbor. Controlled noise is added to these evaluations to guarantee DP. The most promising samples, with duplication allowed, are selected as parents for the next generation. This cycle is repeated for $T$ iterations, progressively guiding the synthetic distribution toward the private data distribution.

While PE is notable as the first customization framework to incorporate DP in black-box settings, it is not free from drawbacks, most notably the loss of diversity. As the evolutionary process proceeds, the ancestry of all samples gradually converges, and the final dataset collapses into a collection of

variations on a single data point. Figure 1 illustrates this issue with a toy example, and Figure 2 presents the image results.

In this paper, we identify the cause of the vanishing diversity problem in PE in the voting method, and propose Diversified Private Evolution (Div-PE), which introduces a two-stage voting scheme to enhance diversity. Inspired by natural ecosystems, where seemingly inferior individuals help maintain genetic variation, we allow samples not selected in the first round to reenter subsequent voting rounds so that they can still contribute to the population. In the second round, superior samples vote for candidates similar to themselves, promoting peer selection. This mechanism prevents superior samples from monopolizing survival, fostering a more diverse and stable synthetic data ecosystem.

While this design intuitively increases diversity, it introduces certain challenges. In particular, treating all samples equally regardless of their fitness can lead to inefficiency. To address this, we allow samples to adaptively vary in the next generation according to their relative distance to the private distribution, assigning inferior samples a larger degree of variation so that they can catch up with superior ones. We also incorporate demonstration-based variation through prompt engineering (Chen et al., 2023; Li & Liang, 2021; Dong et al., 2022b), enabling superior samples to guide inferior ones. In this way, synthetic data samples do not merely compete but support each other in evolving toward their optimal form. Since both the synthetic data and the voting mechanism satisfy DP, this process adheres to privacy standards through the post-processing property (Dwork et al., 2014b), improving data diversity and utility without incurring any additional privacy cost.

By improving diversity in this way, Div-PE achieves greater practical modality scalability than PE. Although PE is theoretically adaptable to various modalities, it has focused primarily on image data. Images lie in a continuous space, whereas text is discrete and tabular data combines both structures. This makes distinguishing variations between samples more difficult in text and tabular domains, increasing the risk of diversity collapse. While Aug-PE (Xie et al., 2024) extends PE to text, tabular data has not yet been explored. Div-PE addresses this gap by incorporating tabular data generation and evaluation, thereby extending synthetic data generation beyond image and text to support a wider range of practical applications.

Our main contributions are summarized as follows:

1. **Diversity**. We propose Div-PE, a framework for differentially private synthetic data generation that achieves substantial diversity improvement in black-box settings.

2. **Quality**. Our method attains SOTA performance with FID $= 48.448$ on ImageNet (Deng et al., 2009), yielding more than an $11\%$ gain in downstream accuracy.

3. **Experimentation**. We provide extensive experiments demonstrating the effectiveness of Div-PE across image, text, and tabular modalities.

## 2 BACKGROUND AND RELATED WORKS

**Differential Privacy (DP).** DP provides a formal way to limit the influence of any single data point on the output of a mechanism (Dwork, 2006). For any two datasets $\mathcal{D}$ and $\mathcal{D}'$ that differ by at most one individual, the output of a mechanism $\mathcal{M}$ remains nearly unchanged so that it is impossible to determine whether a particular record is included in the input.

$$\mathbb{P}\big(\mathcal{M}(\mathcal{D}) \in \mathcal{O}\big) \leq e^\epsilon \mathbb{P}\big(\mathcal{M}(\mathcal{D}') \in \mathcal{O}\big) + \delta \tag{1}$$

If Equation 1 holds for every subset $\mathcal{O}$ of possible outputs of $\mathcal{M}$, the mechanism $\mathcal{M}$ is said to satisfy $(\epsilon, \delta)$-DP.

**Private Evolution (PE).** PE is a framework inspired by evolutionary algorithms (Holland, 1975) that applies DP to image generation models *without* fine-tuning, relying only on black-box access (Lin et al., 2023). Privacy is guaranteed in two key ways. First, throughout the entire process of synthetic data generation, the foundation model never accesses the original data. The procedure begins by generating an initial set of synthetic samples using descriptive prompts. The private data participate only by voting for the synthetic samples that are closest to them in the population, thereby guiding the refinement process. Because the information from private data is reflected solely through voting, this mechanism not only determines the utility of the synthetic data but also represents a potential point of privacy leakage (Hou et al., 2023; Gopi et al., 2020; Hong et al., 2022; Yu et al.,

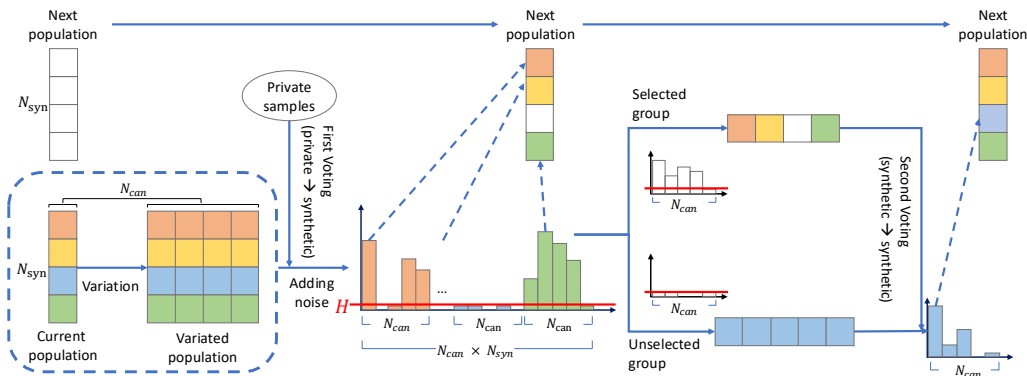

Figure 3: Overview of the proposed Div-PE framework. At each iteration $t$, every sample generates $N_{\text{can}} - 1$ variants to form a candidate set. In the *first voting* (private→synthetic), private samples vote for the nearest candidate, with Gaussian noise $\sigma$ and threshold $H$ applied. Groups passing the threshold advance to the next generation $S_t$, while the others undergo a *second voting* (synthetic→synthetic), where first-stage winners re-vote within each group to ensure at least one survivor. Repeating this process for $T$ iterations yields the final synthetic set $S_T$.

2023). To mitigate this risk, DP is incorporated into the voting mechanism by adding controlled noise to the votes, preventing the exposure of individual records in the original data. However, PE suffers from limited diversity, as repeated iterations favor a small subset of high-performing samples and generate future populations primarily from their variations, leading to homogenization over time.

**Augmented PE (Aug-PE).** Aug-PE (Xie et al., 2024) was proposed to alleviate this limitation by enlarging the intermediate synthetic population by a factor of $L$, assigning a small probability for relatively inferior samples to participate in forming the next generation. However, superior samples still have a higher chance of reproducing multiple offspring, leaving the fundamental issue unresolved. The limited genetic pool of both PE and Aug-PE constrains diversity, which can in turn degrade the performance of downstream tasks trained on the resulting synthetic data (Shipard et al., 2023; Gong et al., 2019; Zhang et al., 2024).

## 3 PROPOSED METHOD

### 3.1 ARCHITECTURE

The proposed framework Div-PE is a synthetic data generation method that approximates the private data distribution by repeatedly performing candidate generation and two-stage voting without any parameter training. The key idea is to (i) generate diverse candidates through prompt-based initialization and variation, and (ii) select the next generation using the two-stage DP voting module BISTAGE to guarantee both diversity and privacy. The overall architecture is illustrated in Figure 3, and the detailed procedures are formalized in Algorithm 1 and Algorithm 2.

**Div-PE Design.** Algorithm 1 formalizes the complete pipeline of Div-PE for a single class. First, a diverse prompt set $P_{\text{pub}}$ is generated using public information $I_{\text{pub}}$ (Line 1, Alg.1). The SEED_API then produces the initial synthetic set $S_0$ (Line 2, Alg.1). At each iteration $t = 1, \ldots, T$, the variation degree $v_t$ is determined by a scheduler $\deg(t)$ (Line 3, Alg.1). From the previous generation $S_{t-1}$, $N_{\text{can}} - 1$ variations are generated to construct the candidate set $S_{\text{can}}^t$ (Lines 6–8, Alg.1). The previous generation samples are then added to the candidate set (Line 11, Alg.1). Finally, the input $(S_{\text{priv}}, S_{\text{can}}^t, \sigma, H, N_{\text{can}})$ is passed to the BISTAGE module (Lines 12–13, Alg.1) to select the next generation $S_t$ through two-stage voting. Repeating this process $T$ times yields the final synthetic set $S_T$.

**Algorithm 1** Diversified Private Evolution (Div-PE)

---

**Input**: Private samples $S_{\text{priv}} = \{x_i^{\text{priv}}\}_{i=1}^{N_{\text{priv}}}$
      Public information $I_{\text{pub}}$
      Number of synthetic samples $N_{\text{syn}}$
      Number of iterations $T$
      Number of candidates $N_{\text{can}}$
      Noise multiplier $\sigma$
      Threshold $H$
      Variation scheduler $\deg(\cdot)$
**Output**: Synthetic samples $S_T$

1:  $P_{\text{pub}} \leftarrow \text{Prompt\_Generate}(N_{\text{syn}}, I_{\text{pub}})$
2:  $S_0 \leftarrow \text{SEED\_API}(P_{\text{pub}})$
3:  **for** $t \leftarrow 1, \ldots, T$ **do**
4:    $v_t \leftarrow \deg(t)$
5:    $S_{\text{can}}^t \leftarrow \emptyset$
6:    **for** $s_i \in S_{t-1}$ **do**
7:      **for** $c \leftarrow 1, \ldots, N_{\text{can}} - 1$ **do**
8:        $z \leftarrow \text{VARIATE\_API}(s_i, v_t)$
9:        $S_{\text{can}}^t \leftarrow S_{\text{can}}^t \cup \{z\}$
10:     **end for**
11:   **end for**
12:   $S_{\text{can}}^t \leftarrow S_{\text{can}}^t \cup S_{t-1}$
13:   $inp \leftarrow (S_{\text{priv}}, S_{\text{can}}^t, \sigma, H, N_{\text{can}})$
14:   $S_t \leftarrow \text{BISTAGE}(inp)$
15: **end for**
16: **return** $S_T$

**Algorithm 2** BI-stage Voting (BISTAGE)

---

**Input**: Private samples $S_{\text{priv}}$
      Candidate pool $S_{\text{can}} = \{z_j\}_{j=1}^{N_{\text{can}} \times N_{\text{syn}}}$
      Number of candidates $N_{\text{can}}$
      Noise multiplier $\sigma$
      Threshold $H$
      Distance function $d(\cdot, \cdot)$
**Output**: Selected samples $S$

1:  $V^{(1)} \leftarrow [0, \ldots, 0]$
2:  **for** $x_i^{\text{priv}} \in S_{\text{priv}}$ **do**
3:    $\delta_j \leftarrow d(\Phi(x_i^{\text{priv}}), \Phi(z_j))$
4:    $j^* \leftarrow \arg\min_j \delta_j$
5:    $V^{(1)}[j^*] \leftarrow V^{(1)}[j^*] + 1$
6:  **end for**
7:  $V^{(1)} \leftarrow V^{(1)} + \mathcal{N}(0, \sigma^2 I)$
8:  $V^{(1)} \leftarrow \max(V^{(1)} - H, 0)$
9:  $best^{(1)} \leftarrow \text{FIND\_BEST}(V^{(1)}, N_{\text{can}})$
10:  $(\text{idx\_sel}^{(1)}, \text{idx\_uns}^{(1)}) \leftarrow best^{(1)}$
11:  $V^{(2)} \leftarrow [0, \ldots, 0]$
12:  **for** $s \in \text{idx\_sel}^{(1)}$ **do**
13:    **for** $u \in \text{idx\_uns}^{(1)}$ **do**
14:      $\delta_j \leftarrow d(\Phi(S_{\text{syn}}[s]), \Phi(z_j))$
15:      $j^* \leftarrow \arg\min_j \delta_j$
16:      $V^{(2)}[j^*] \leftarrow V^{(2)}[j^*] + 1$
17:    **end for**
18:  **end for**
19:  $best^{(2)} \leftarrow \text{FIND\_BEST}(V^{(2)}, N_{\text{can}})$
20:  $(\text{idx\_sel}^{(2)}, \_) \leftarrow best^{(2)}$
21:  $S \leftarrow S_{\text{syn}}[\text{idx\_sel}^{(1)} \cup \text{idx\_sel}^{(2)}]$
22: **return** $S$

**Two-Stage Voting Mechanism.** Algorithm 2 details the BISTAGE module that selects the next generation from the candidate set $S_{\text{can}}$. In the *First Voting* stage, each private sample $x_i^{\text{priv}}$ votes for its nearest candidate $z_{j^*}$ in the embedding space (Lines 1–5, Alg.2). Gaussian noise $\sigma$ is then added to the vote vector $V^{(1)}$, and a threshold $H$ is applied (Lines 6–8, Alg.2). The FIND\_BEST function selects the top-voted candidate in each group (Line 9, Alg.2). The detailed selection procedure is described in Appendix A. Groups not selected in the first round proceed to the *Second Voting* stage, where first-round winners vote within their own groups without noise to finalize the selection (Lines 10–18, Alg.2). This process ensures that each group contributes at least one candidate to the next generation $S_t$. For multi-class settings, the procedure is executed independently for each class.

**Auto-Prompt.** To broaden the diversity of synthetic data generation, Div-PE employs an Auto-Prompt strategy that automatically expands the prompt set based on public information $I_{\text{pub}}$ using large language models (LLMs). We assume that $I_{\text{pub}}$ consists of only a single keyword or short phrase (*e.g.* a photo of a cat), which provides a limited expressive range and restricts both the diversity and the convergence speed of the candidate set $S_{\text{can}}^t$. To mitigate this, an LLM is used to generate a prompt set

$$P_{\text{pub}} = \{p_1, p_2, \ldots, p_{N_{\text{syn}}}\}$$

that matches the target number of synthetic samples $N_{\text{syn}}$ by enriching each input concept with detailed attributes and conditions. Specifically, the following instruction is applied to guide the LLM to produce richer yet single-sentence prompts:

> Make the following prompt more descriptive by adding appropriate details,
> but end the result naturally with a single sentence.
> {Input Prompt}
> Enhanced Prompt:

This template converts each {Input Prompt} into an enhanced single-sentence prompt containing contextual and concrete attributes.

**Demonstration-Based Variation.** For generations $t \geq 2$, superior candidates guide the variation of inferior candidates. Each synthetic sample $z_i \in S_{t-1}$ forms a demonstration set $D_i^t$ by probabilistically selecting higher-voted samples according to the noise-added first voting vector $V^{(1,t-1)}$:

$$P(z_j \in D_i^t) = \frac{V_j^{(1,t-1)}}{\sum_{k \in X_i^t} V_k^{(1,t-1)}}, \quad X_i^t = \left\{ j \mid V_j^{(1,t-1)} \geq V_i^{(1,t-1)} \right\}. \tag{2}$$

Since $V^{(1,t-1)}$ already includes DP noise from Algorithm 2 (Lines 7–8), no additional privacy cost is incurred.

**Adaptive Variation.** Synthetic samples farther from the private distribution are assigned a higher degree of variation to accelerate convergence. For each sample $z_i \in S_{t-1}$, the variation degree $v_i^t$ at generation $t$ is set as

$$v_i^t = \deg(t) \times \max\left( 0.1, \ 1 - \frac{V_i^{(1,t-1)}}{N_{\text{priv}}} \right) \tag{3}$$

where $V_i^{(1,t-1)}$ is the noise-added first voting count. A smaller $V_i^{(1,t-1)}$ implies greater distance from the private distribution and thus allows a larger variation. The scheduler $\deg(t)$ controls the global exploration strength and can be increased when high-vote candidates remain distant from the private distribution, which is typically the case in the early stages.

## 3.2 RATIONALE AND VALIDATION

### 3.2.1 DIFFERENTIAL PRIVACY

In the first voting of the $t$-th BISTAGE, the noise-added votes $x_i \in S_{\text{can}}^t$ received from private samples are defined as

$$V_i^t = \max\big( f(x_i) + \mathcal{N}(0, \sigma^2 \mathrm{I}) - H, 0 \big) \tag{4}$$

where $\mathcal{N}(0, \sigma^2 \mathrm{I})$ denotes Gaussian noise for privacy protection. Each private sample contributes only one vote. Changing a single instance in $S_{\text{priv}}$ changes the first voting result by at most 1 in the $l_2$ norm, giving sensitivity 1. The post-processing property guarantees that any further processing of DP-protected data remains DP. Because the first voting stage satisfies $(\epsilon, \delta)$-DP, the second stage also satisfies $(\epsilon, \delta)$-DP. DP is applied only once per iteration, so applying the Gaussian mechanism (Dwork et al., 2014a) over $T$ iterations follows $T$ adaptive composition (Dong et al., 2022a), as analyzed in (Lin et al., 2023). Thus each iteration consumes a privacy budget of $\sigma/\sqrt{T}$ within the total budget $\epsilon$.

**Why not select many samples in the first voting.** Each original sample selects a single synthetic sample, keeping DP sensitivity at 1. Allowing multiple selections would increase sensitivity, require more noise, and reduce the utility of synthetic data.

**Why use synthetic data in the second voting.** The privacy budget analysis assumes exactly one private-data-based vote per iteration. If private data voted more than once per iteration, the privacy budget would increase proportionally. Div-PE improves diversity without additional privacy cost by using synthetic data for the second voting.

### 3.2.2 ENSURING CONVERGENCE

Following the assumption of PE that at least $M \gg H$ private points lie in an $L_2$ ball of diameter $D$ where the SEED_API generates initial samples, private points in each cluster tend to vote for the same synthetic sample when $S_{\mathrm{syn}}$ is far from the private distribution. The selected samples converge to $S_{\mathrm{prv}}$ within Wasserstein distance $\leq \eta$ ($\forall p \in [1, \infty]$) with probability $\geq 1 - \tau$ whenever

$$T \gg \frac{d \log(D/\eta)}{\log N_{\mathrm{can}}} + \log(N_{\mathrm{prv}}/\tau), \tag{5}$$

where $d$ is the intrinsic dimension of the embedding space. Because the second-round voting already satisfies DP, its convergence proof follows that of the non-private case (Lin et al., 2023).

**Why select only one candidate per group in the second voting.** The main limitation of PE arises from the functional gap between the SEED_API and the VARIATE_API. The VARIATE_API produces only minor variations around selected samples and cannot match the diversity of the SEED_API. If the VARIATE_API provided diversity comparable to the SEED_API, the voting-based convergence principle of PE would fail. Therefore, leveraging the diversity of the SEED_API requires selecting exactly one candidate per group to maintain diversity.

### 3.2.3 COST EFFICIENCY

Div-PE differs from PE only in the second voting. Let $N_{\mathrm{syn}}$ be the number of synthetic samples per iteration and $N_{\mathrm{sel}}$ the subset chosen in the first stage. The second stage selects the remaining $N_{\mathrm{syn}} - N_{\mathrm{sel}}$ samples from unselected groups.

1. Selecting $N_{\mathrm{sel}}$ samples from $N_{\mathrm{syn}}$ candidates requires $\mathcal{O}(N_{\mathrm{sel}})$ time.
2. Using Faiss, the nearest neighbor search takes $\mathcal{O}(\log N_{\mathrm{syn}})$ for each of the $N_{\mathrm{sel}}$ samples, giving a total complexity of $\mathcal{O}(N_{\mathrm{sel}} \cdot \log N_{\mathrm{syn}})$.

Since the linear term dominates, the overall complexity of the second voting phase is

$$\mathcal{O}(N_{\mathrm{sel}} \cdot \log N_{\mathrm{syn}}). \tag{6}$$

Moreover, Div-PE calls APIs only during candidate generation, while the second voting operates on pre-generated candidates. Parameters affecting API usage (*e.g.* $K$, $L$, and $N_{\mathrm{can}}$) are fixed across experiments, ensuring identical API consumption for PE, Aug-PE, and Div-PE. Under these conditions, Div-PE achieves superior performance with equal cost.

**Why not increase variations of superior samples to improve diversity.** Increasing the number of variations requires additional API calls, which incur monetary cost for commercial black-box APIs or GPU cost for local execution. Aug-PE improves diversity by generating more variations, whereas Div-PE achieves better diversity at lower cost by enhancing the voting mechanism.

## 4 EXPERIMENTS

**Data.** We evaluate the proposed method across three modalities using representative benchmarks. For image data we adopt ImageNet (Deng et al., 2009), Camelyon17 (Litjens et al., 2018), and UTKFace (Zhang et al., 2017). For text data we use OpenReview (Xie et al., 2024) with acceptance labels and Yelp (Zhang et al., 2015) with star ratings. For tabular data we use Adult (Becker & Kohavi, 1996) with binary income labels and Body-Performance (Cho & contributors, 2021) with multi-class physical condition labels.

**Models.** Our framework relies on two unified interfaces: a SEED_API for initial generation and a VARIATE_API for controlled refinement. Stable Diffusion v1.5 (Rombach et al., 2022) is used as the image generator, while Llama-2-7b-hf (Touvron et al., 2023) serves as the backbone for text and tabular generation as well as for auto-prompt and variation across modalities. Further implementation details are provided in the Appendix B.

| Modality | Dataset | Method | ACC (↑) | FID/W-dist (↓) | Precision (↑) | Recall (↑) | Density (↑) | Coverage (↑) |
|---|---|---|---|---|---|---|---|---|
| Image | ImageNet | PE | 0.411 | 81.204 | 0.934 | 0.000 | 0.998 | 0.194 |
| | | Aug-PE | 0.732 | 48.359 | 0.891 | 0.677 | 0.998 | 0.545 |
| | | Ours | **0.889** | **45.058** | **0.878** | 0.247 | **0.924** | **0.648** |
| | Camelyon17 | PE | 0.626 | 290.747 | 0.000 | 0.000 | 0.000 | 0.000 |
| | | Aug-PE | 0.814 | 214.704 | 0.009 | 0.277 | 0.001 | 0.004 |
| | | Ours | **0.861** | **187.404** | 0.014 | **0.175** | **0.002** | **0.004** |
| | UTKFace | PE | 0.616 | 246.015 | 0.182 | 0.000 | 0.021 | 0.003 |
| | | Aug-PE | 0.732 | 167.086 | 0.082 | 0.094 | 0.011 | 0.005 |
| | | Ours | **0.774** | **113.497** | **0.084** | **0.048** | **0.012** | **0.020** |
| Text | OpenReview | Aug-PE | 0.370 | 0.017 | 0.104 | 0.551 | 0.031 | 0.032 |
| | | Ours | **0.440** | **0.012** | **0.233** | **0.625** | **0.163** | **0.254** |
| | Yelp | Aug-PE | 0.620 | 0.017 | 0.106 | **0.696** | 0.031 | 0.025 |
| | | Ours | **0.660** | **0.012** | **0.238** | 0.601 | **0.164** | **0.267** |
| Tabular | Adult | Ours | 0.918 | 0.017 | 0.721 | 0.719 | 0.340 | 0.629 |
| | Body-Performance | Ours | 0.772 | 0.023 | 0.460 | 0.436 | 0.126 | 0.394 |

Table 1: Overall performance across modalities under a fixed DP budget ($\epsilon$=2.0, $\delta$=$10^{-4}$, $T$=17) and candidate breadth. Best values within each dataset are bolded.

**Metrics.** We evaluate two key aspects. (i) *Distributional similarity* is measured by FID for images (Heusel et al., 2017), Wasserstein distance for text and tabular data, and by density and coverage for local and global alignment (Naeem et al., 2020). (ii) *Label-conditioned utility* is measured by downstream accuracy, where models are trained on synthetic data and evaluated on held-out test sets. We use a ResNet-18 classifier for images, a RoBERTa-based classifier for text (aligned with Aug-PE), and a Random Forest classifier for tabular data. Higher density, coverage, and accuracy are desirable (↑), whereas lower FID and Wasserstein indicate better fidelity (↓).

**Hyperparameters.** All methods use $\epsilon = 2.0$, $\delta = 10^{-4}$, and $T = 17$. PE fixes pool size $K = 8$, Aug-PE sets prompt-side exploration $L = 8$, and our two-stage method uses $N_{\text{can}} = 8$. Other hyperparameters are reported in the Appendix B.

## 4.1 MODALITY-SPECIFIC IMPLEMENTATIONS.

Across all modalities we apply two-stage selection, auto-prompt, demonstration and adaptive variation. For tabular data we employ GreaT (Borisov et al., 2022) to serialize rows into natural language (*e.g.*, "column1 is value1, column2 is value2, ..."), which enables schema-preserving text-based processing. For demonstration, images use IP-Adapter conditioning (et al., 2023; Cubiq, 2024), whereas text and tabular data use system-prompt exemplars.

## 4.2 OVERALL GENERATION PERFORMANCE

Table 1 summarizes results across all datasets. On ImageNet, compared to Aug-PE with accuracy 0.732, our method reaches 0.889; coverage expands from 0.545 to 0.648, and FID decreases from 48.359 to 45.058. Unlike PE with recall 0.000, ours maintains balanced recall 0.247 together with strong precision. On Camelyon17, accuracy improves from 0.814 in Aug-PE to 0.861, and FID drops from 214.704 to 187.404. Coverage stays at 0.004 in both, reflecting the limitation of natural-image–pretrained foundation models in histopathology. Here PE collapses to density 0.000, while ours preserves a small but non-trivial density of about 0.002. On UTKFace, accuracy rises from 0.732 in Aug-PE to 0.774, FID falls from 167.086 to 113.497, and coverage grows from 0.005 to 0.020, expanding support beyond the baseline.

For text benchmarks, Aug-PE benefits from prompt expansion but coverage remains limited. On OpenReview, accuracy increases from 0.370 in Aug-PE to 0.440, Wasserstein distance decreases from 0.017 to 0.012, and coverage expands from 0.032 to 0.254. On Yelp, accuracy rises from 0.620 to 0.660, Wasserstein decreases from 0.017 to 0.012, and coverage grows from 0.025 to 0.267, providing more balanced class representation.

| Method Variant (ImageNet) | ACC (↑) | FID (↓) | Precision (↑) | Recall (↑) | Density (↑) | Coverage (↑) |
|---|---|---|---|---|---|---|
| Ours (auto) | 0.830 | 48.448 | 0.795 | 0.301 | 0.860 | 0.674 |
| Ours (auto+av) | 0.850 | 48.475 | 0.783 | 0.418 | 0.871 | 0.657 |
| Ours (auto+demo) | 0.868 | 136.818 | 0.895 | 0.083 | 0.928 | 0.301 |
| Ours (all) | 0.889 | 45.058 | 0.878 | 0.247 | 0.924 | 0.648 |

Table 2: Ablation on ImageNet under a fixed DP budget ($\epsilon$=2.0, $\delta$=$10^{-4}$, $T$=17). **auto**: auto-prompt only; **auto+av**: auto-prompt with adaptive variation; **auto+demo**: auto-prompt with demonstration; **all**: auto-prompt with adaptive variation and demonstration.

For tabular data, our method attains accuracy 0.918 on Adult with density 0.340 and coverage 0.629, and accuracy 0.772 on Body-Performance with density 0.126 and coverage 0.394. This shows that serialization-based generation preserves schema validity while broadening support.

### 4.3 ANALYSIS OF ABLATION PERFORMANCE

Table 2 evaluates auto-prompt, adaptive variation, and demonstration. Auto-prompt alone improves coverage to 0.674 (vs. 0.545 in Aug-PE) with density 0.860. Adding adaptive variation raises density further to 0.871 and coverage 0.657, preventing collapse into dominant modes. Demonstration guidance increases precision to 0.895 but reduces coverage to 0.301, despite high density 0.928. Combining all three yields the most balanced outcome: accuracy 0.889, FID 45.058, density 0.924, and coverage 0.648. In summary, auto-prompt diversifies support, adaptive variation stabilizes density, and demonstration sharpens fidelity, together avoiding the collapse of PE and the instability of Aug-PE. In summary, auto-prompt diversifies support, adaptive variation stabilizes density, and demonstration sharpens fidelity, together avoiding the collapse of PE and the instability of Aug-PE.

## 5 CONCLUSION

We proposed Div-PE, a framework designed to generate DP synthetic data without the need for additional training. This approach is particularly suitable for the growing prevalence of black-box environments, where model parameters remain inaccessible. While similar approaches have been explored previously, Div-PE stands out by employing a two-stage voting mechanism and prompt engineering to enhance diversity and adaptive variation, along with demonstration-based variation, thereby improving the quality of synthetic data for a more diverse synthetic samples.

While Div-PE enables synthetic data generation using only APIs and leverages proprietary models without privacy concerns, it still faces inherent limitations. These limitations primarily arise from the unavoidable replication of biases in foundation models and the dependence on pre-existing knowledge within the model. This becomes particularly challenging in fields requiring high levels of expertise, such as medical data, where the absence of model training can result in significant deficiencies. This issue reflects ongoing research trends, which aim to control model outputs without additional training (Dekoninck et al., 2024; Wang et al., 2024). In addition, there exists a trade-off between diversity and convergence speed: drawing from a wider range of groups improves diversity but slows convergence, whereas prior methods such as PE achieved faster convergence at the cost of reduced diversity. Addressing these concerns will be crucial in the ongoing development of synthetic data methodologies.

## 6 ETHICS STATEMENT

Our algorithm is designed to employ foundation models in a black-box manner while protecting the privacy of the target dataset. Accordingly, the safety of the foundation models themselves is beyond the scope of this study. Nevertheless, foundation models may exhibit biases inherited from their training data, and they remain susceptible to prompt injection and related attacks that can bypass built-in safeguards, potentially resulting in the generation of harmful, illegal, or sensitive content. We emphasize that these risks originate from the underlying models rather than from our proposed framework. Addressing such safety challenges remains an important direction for future research in responsible deployment of large-scale generative models.

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

# APPENDIX

## A  SELECTING BEST SAMPLES

---

**Algorithm 3** Find the Best Candidate for Each Group (FIND_BEST)

---

**Input**: Vote vector $votes$
      Number of candidates per group $N_{\text{can}}$
**Output**: Best indices $idx\_best$, Zero-vote group indices $idx\_zero$

1: $n \leftarrow \lceil |votes|/N_{\text{can}} \rceil$
2: $idx\_best \leftarrow \emptyset$, $idx\_zero \leftarrow \emptyset$
3: **for** $g \leftarrow 0, \ldots, n-1$ **do**
4:    $start \leftarrow g \times N_{\text{can}}$
5:    $end \leftarrow \min\big((g+1) \times N_{\text{can}}, |votes|\big)$
6:    $b \leftarrow \arg\max_{j \in [start, end)} votes[j]$
7:    **if** $votes[b] = 0$ **then**
8:       $idx\_zero \leftarrow idx\_zero \cup \{start\}$
9:    **else**
10:      $idx\_best \leftarrow idx\_best \cup \{b\}$
11:   **end if**
12: **end for**
13: **return** $idx\_best,\ idx\_zero$

---

The function takes the following inputs:

- **Votes:** An array containing the vote totals for each candidate.
- **Number of Candidates** ($N_{\text{can}}$)**:** The number of variations plus the original sample, forming a group.

And produces the following outputs:

- **BestIndex:** Indices of the candidates with the highest votes in each group.
- **ZeroIndex:** Indices of the groups where all members received zero votes.

## B  EXPERIMENT SETTINGS

### B.1  HYPERPARAMETERS

Table 4 summarizes the hyperparameters used in the experiments for PE, Aug-PE, and Div-PE. This includes both shared parameters and method-specific settings, providing a comprehensive overview necessary for reproducing the experiments and understanding the model performance.

The degree scheduler adjusts the variation degree over $T$, linearly decreasing it from degree_scheduler_base to degree_scheduler_min. The num_candidate parameter specifies the number of candidates considered in each iteration. The prompt generator refers to the model used to extract public prompts from public data. The demonstration parameter indicates the maximum number of demonstration samples that can be utilized. In adaptive variation, the variation degree is linearly adjusted from weight_scheduler_base to weight_scheduler_min over $T$ iterations.

### B.2  HARDWARE SPECIFICATION

Table 4 summarizes the hardware specification of the system where every experiment were conducted in the paper. We have 4 GPUs in total but omitted the number since only a single GPU per experiment was utilized.

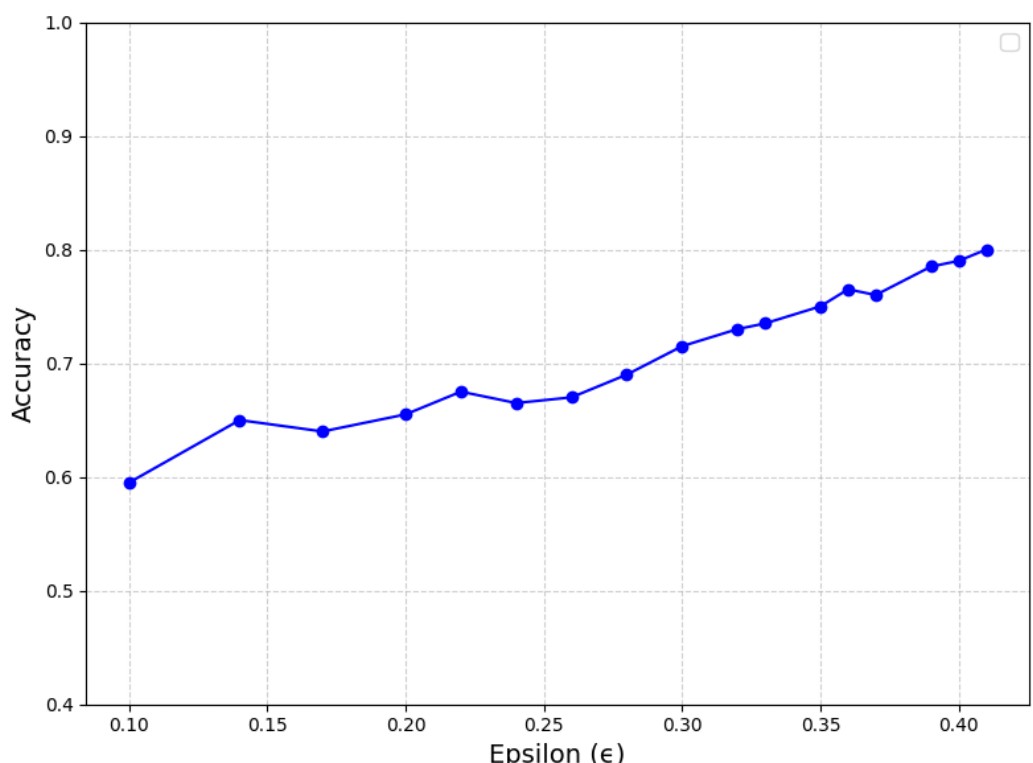

Figure 4: The graph illustrates the accuracy trends of downstream classification accuracy using synthetic data generated for the ImageNet Goldfinch and Indigo Bunting classes through our DPSDivA algorithm. The synthetic data was created with $\epsilon$ values ranging from 0.1 to 0.41, divided into 17 equal intervals. The results demonstrate the impact of varying $\epsilon$ on the downstream classification accuracy, highlighting the relationship between differential privacy settings and classification accuracy. Notably, when $\epsilon \geq 0.35$, the accuracy exceeds 0.70, and even at very low $\epsilon = 0.41$, an accuracy of over 0.80 is achieved, approaching the original data's accuracy of 0.85.

| Parameter | PE | Aug-PE | Div-PE |
|---|---|---|---|
| count_threshold | | 2 | |
| degree_scheduler | | linear | |
| degree_scheduler_base | | 1 | |
| degree_scheduler_min | | 0.7 | |
| T | | 17 | |
| feature extractor | | clip_vit_b_32 | |
| model | | stable-diffusion-v1-5 | |
| guidance scale | | 7.5 | |
| number of steps | | 20 | |
| K | 8 | 1 | - |
| L | - | 8 | - |
| num_candidate | - | - | 8 |
| demonstration | - | - | 3 |
| weight_scheduler_min | - | - | 0.8 |
| weight_scheduler_base | - | - | 1 |

Table 3: Comparison of hyperparameters for PE, Aug-PE, and Div-PE. Shared values are centered, and a separator line is added above method-specific parameters.

| | |
|---|---|
| GPU | NVIDIA RTX A6000 |
| Memory | 8 M393A8G40BB4-CWE |
| System | SYS-740GP-TNRT |
| Processor | 112 Intel(R) Xeon(R) Gold 6348 CPU @ 2.60GHz |
| OS | Ubuntu 18.04 |

Table 4: Hardware specification

## C  PERFORMANCE ANALYSIS

### C.1  EFFECTS OF INDIVIDUAL COMPONENTS

Figure 6 illustrates the impact of the detailed components of Div-PE on KID and coverage. Div-PE outperforms both PE and Aug-PE in both metrics. A closer examination of its individual components reveals that applying all elements of Div-PE yields the best performance. Between the demonstration-based variation and adative variation, the former achieves better results in terms of KID, while the latter excels in coverage. It is important to note that KID reflects distributional similarity, whereas coverage measures diversity.

The demonstration-based variation promotes the generation of samples resembling superior examples by directly influencing other samples, which enhances distributional similarity but slightly reduces diversity. In contrast, the adaptive variation assigns higher variation degrees to inferior samples, allowing for more freedom in their transformation. This approach benefits diversity but may compromise distributional similarity. The best performance observed when both components are applied simultaneously suggests that they have complementary effects.

### C.2  EFFECTS OF PRIVACY PARAMETERS

Figure 4 presents the accuracy trends observed in downstream classification accuracy using synthetic data generated for the ImageNet Goldfinch and Indigo Bunting classes through our proposed DPSDivA algorithm. The synthetic data was generated with privacy budgets ($\epsilon$) ranging from 0.1 to 0.41, divided into 17 equal intervals. The figure illustrates the relationship between differential privacy parameters and downstream classification accuracy, showcasing the trade-off between privacy and performance. Specifically, when $\epsilon \geq 0.35$, the classification accuracy surpasses 0.70, and even at a relatively low privacy budget of $\epsilon = 0.41$, the accuracy exceeds 0.80, closely approaching the original data's accuracy of 0.85. This demonstrates the effectiveness of the DPSDivA algorithm in maintaining high utility under strict privacy constraints.

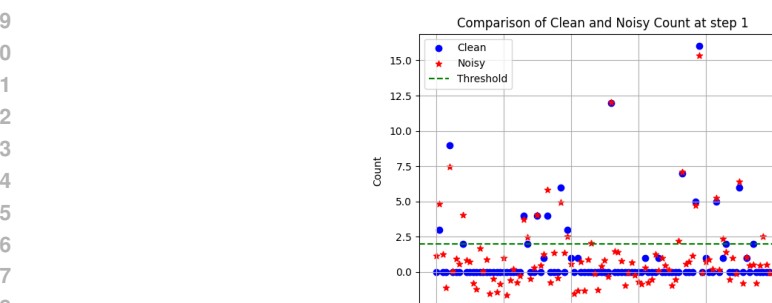

Figure 5: The distribution of clean and noisy counts in the first variation.

| Sample Index | Proportion (%) |
|:---:|:---:|
| 73 | 15.0 |
| 68 | 12.2 |
| 61 | 8.8 |
| 59 | 7.2 |
| 55 | 6.4 |
| 97 | 5.1 |
| 37 | 4.9 |
| 51 | 3.7 |

Table 5: The proportion of noisy counts for samples exceeding the threshold.

## D  DIVERSITY ANALYSIS

Figure 5 shows the results of the voting for the first variation. It is important to note that the point at which our algorithm clearly diverges from the baseline is during the voting and variations that follow the initial generation. As shown in the figure, in the initial population, most samples do not receive votes, with only a few minority samples receiving votes. Based on these results, if the next variation is conducted, in the case of PE or AUG-PE, approximately 15 out of 100 samples would become variations of sample 73 (Table 5). Specifically, out of 100 samples, 82 received a score of 0, and only 18 samples received all the votes. In such cases, both PE and AUG-PE eliminate the samples that did not receive votes immediately, whereas our algorithm does not, allowing it to maintain greater diversity compared to the two algorithms.

Figure 7 presents the PCA analysis for each algorithm. PCA reduction was performed independently at each step rather than being fixed throughout. Notably, while the synthetic distributions of the three algorithms appear similar at step 0, they exhibit significant differences at step 17. It is also evident that the synthetic data at step 0 already demonstrates reduced diversity compared to the original data, highlighting a fundamental limitation of synthetic data generation.

However, the diversity issue becomes even more pronounced for PE and Aug-PE, as their evolutionary processes lead to a further reduction in diversity. In contrast, our algorithm successfully maintains the initial diversity observed at step 0, demonstrating its robustness in preserving data variation.

## E  MORE IMAGES ON IMAGENET

To highlight the limitations of existing methods in terms of diversity and demonstrate the effectiveness of our proposed approach, we present additional generation results on ImageNet. As shown in Figure 8 and 12, the original data exhibits a high degree of diversity, whereas the results generated by PE (Figure 9 and 13) and AUG-PE (Figure 10 and 14) lack such diversity. Figure 11 and

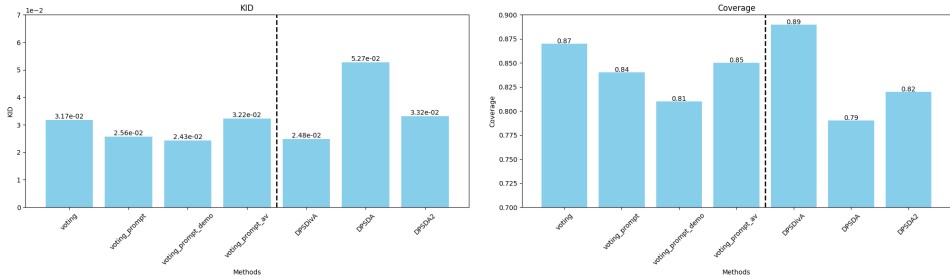

Figure 6: We calculated KID (top) and coverage (bottom) by incorporating each component of Div-PE — Auto-Prompt (prompt), demonstration-based variation (demo), and adaptive variation (av) on BISTAGE (voting) — and compared the results with the baselines.

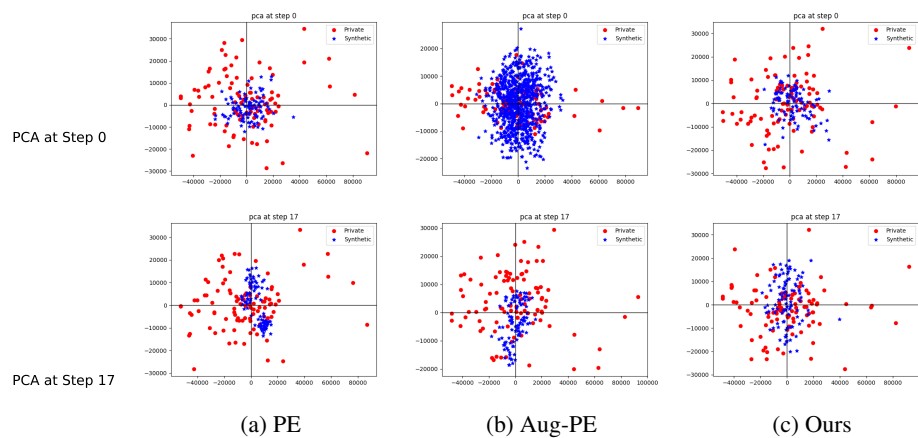

(a) PE          (b) Aug-PE          (c) Ours

Figure 7: The PCA visualizations clearly illustrate the disparity in sample diversity among the algorithms.

15 demonstrate higher diversity compared to both baselines. These results directly demonstrate the effectiveness of Div-PE.

## F  USE OF LARGE LANGUAGE MODELS

Large language models (e.g., ChatGPT) were used only for ancillary tasks, such as language editing and translation of draft text. They did not contribute to the conception of ideas, experiment design, or writing of substantive scientific content.

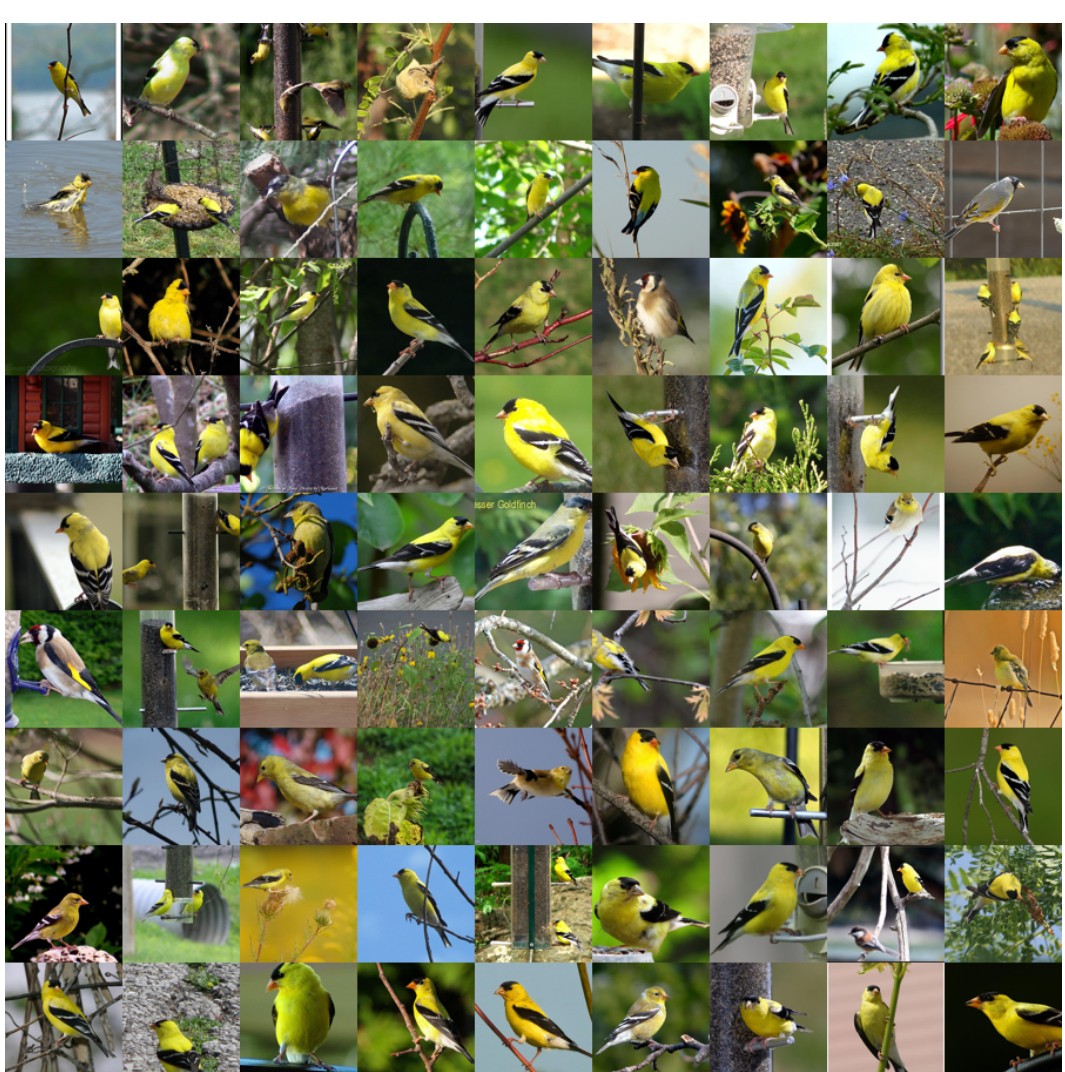

Figure 8: Original data from the Goldfinch class in the ImageNet dataset.

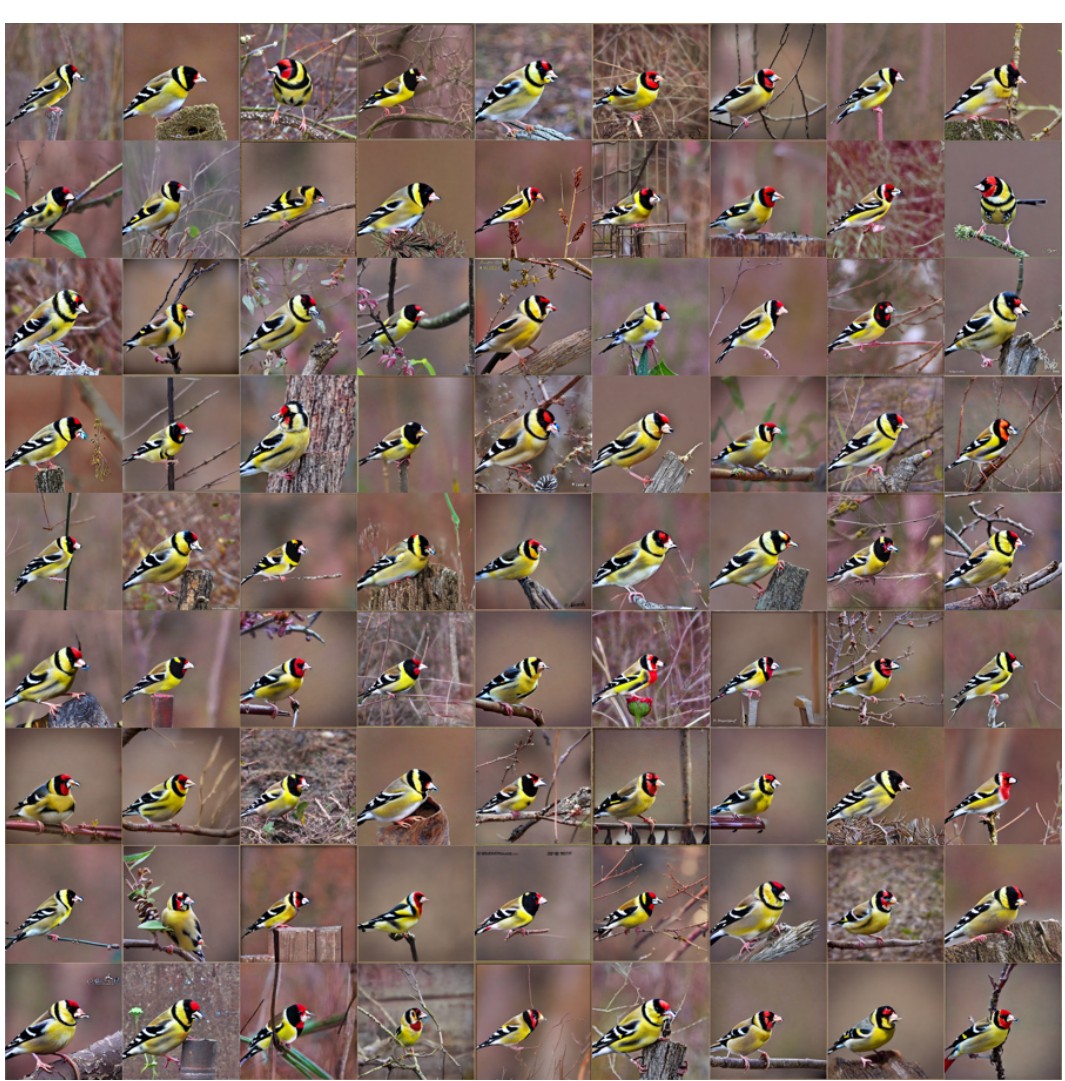

Figure 9: Synthetic data generated for the Goldfinch class in the ImageNet dataset using the DPSDA.

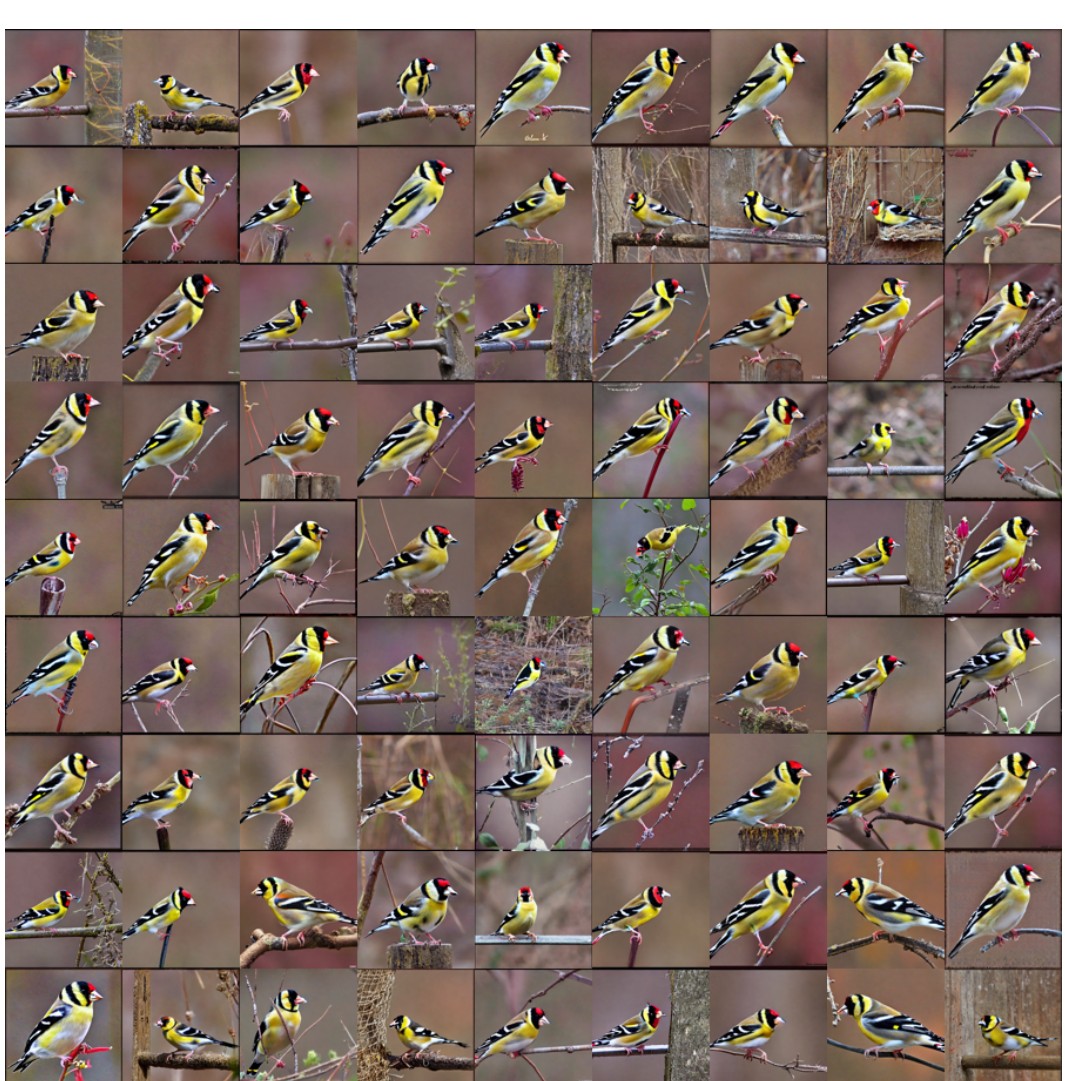

Figure 10: Synthetic data generated for the Goldfinch class in the ImageNet dataset using the DPSDA2.

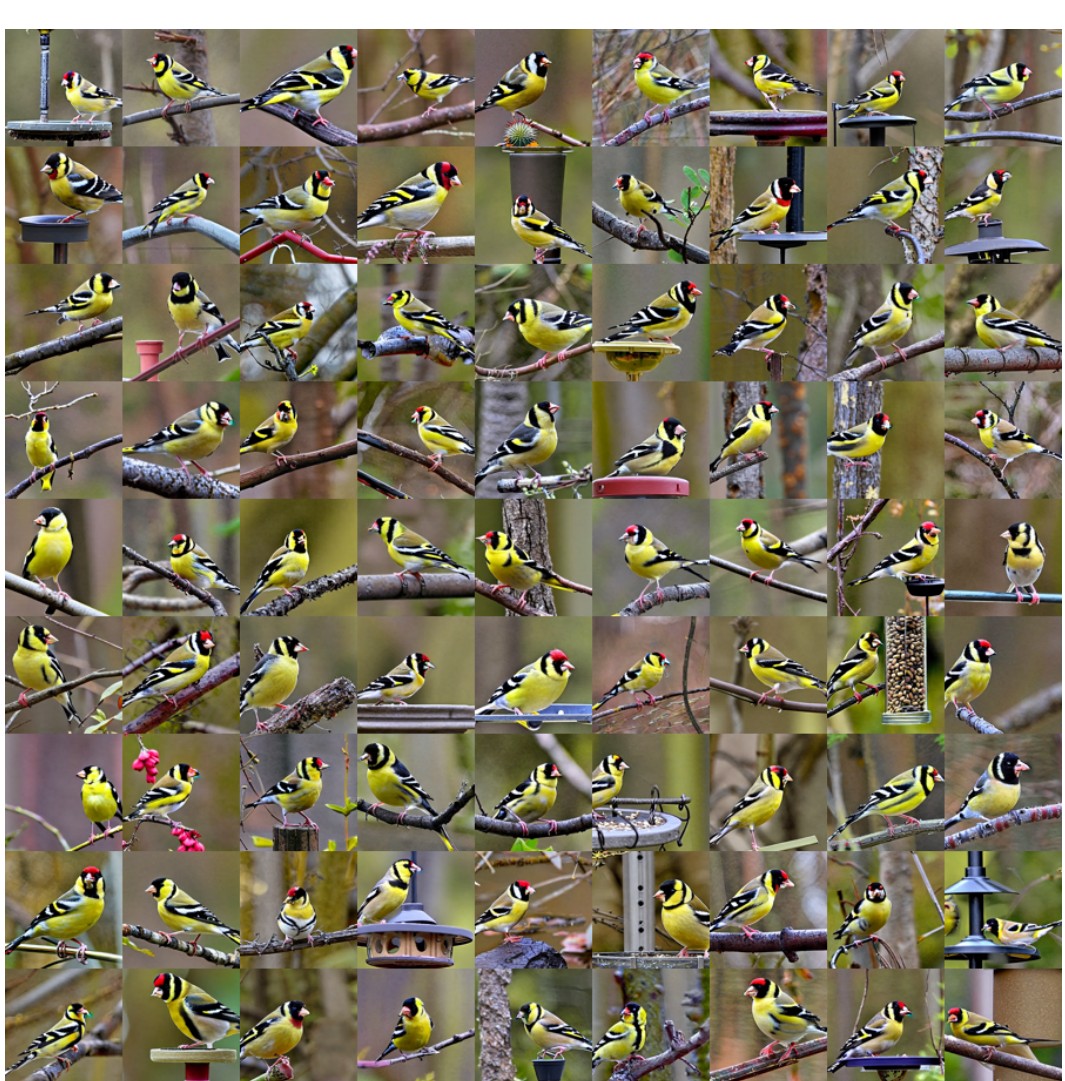

Figure 11: Synthetic data generated for the Goldfinch class in the ImageNet dataset using the DPS-DivA.

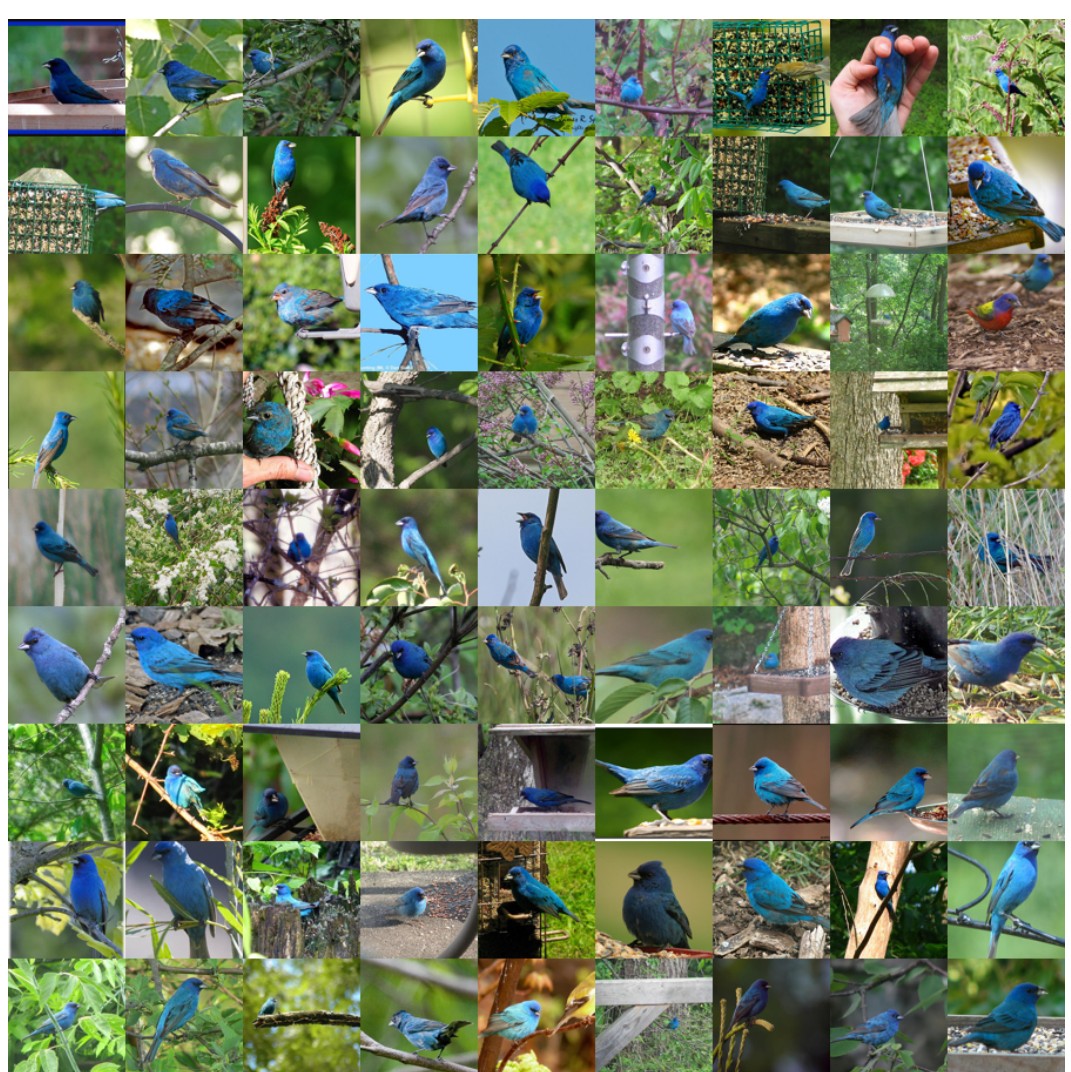

Figure 12: Original data from the Indigo Bunting class in the ImageNet dataset.

1296
1297
1298
1299
1300
1301
1302
1303
1304
1305
1306
1307
1308
1309
1310
1311
1312
1313
1314
1315
1316
1317
1318
1319
1320
1321
1322
1323
1324
1325
1326
1327
1328
1329
1330
1331
1332
1333
1334
1335
1336
1337
1338
1339
1340
1341
1342
1343
1344
1345
1346
1347
1348
1349

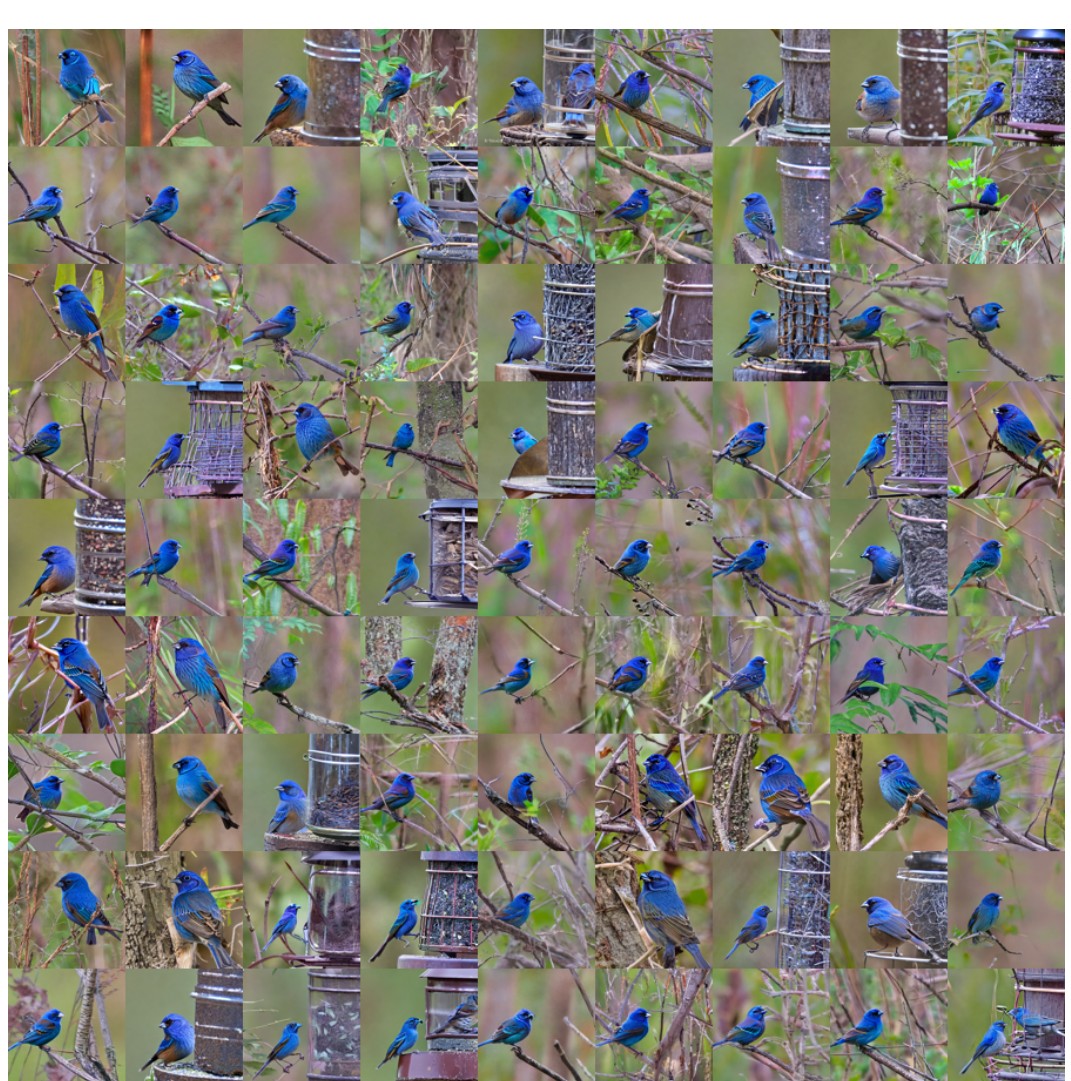

Figure 13: Synthetic data generated for the Indigo Bunting class in the ImageNet dataset using the DPSDA.

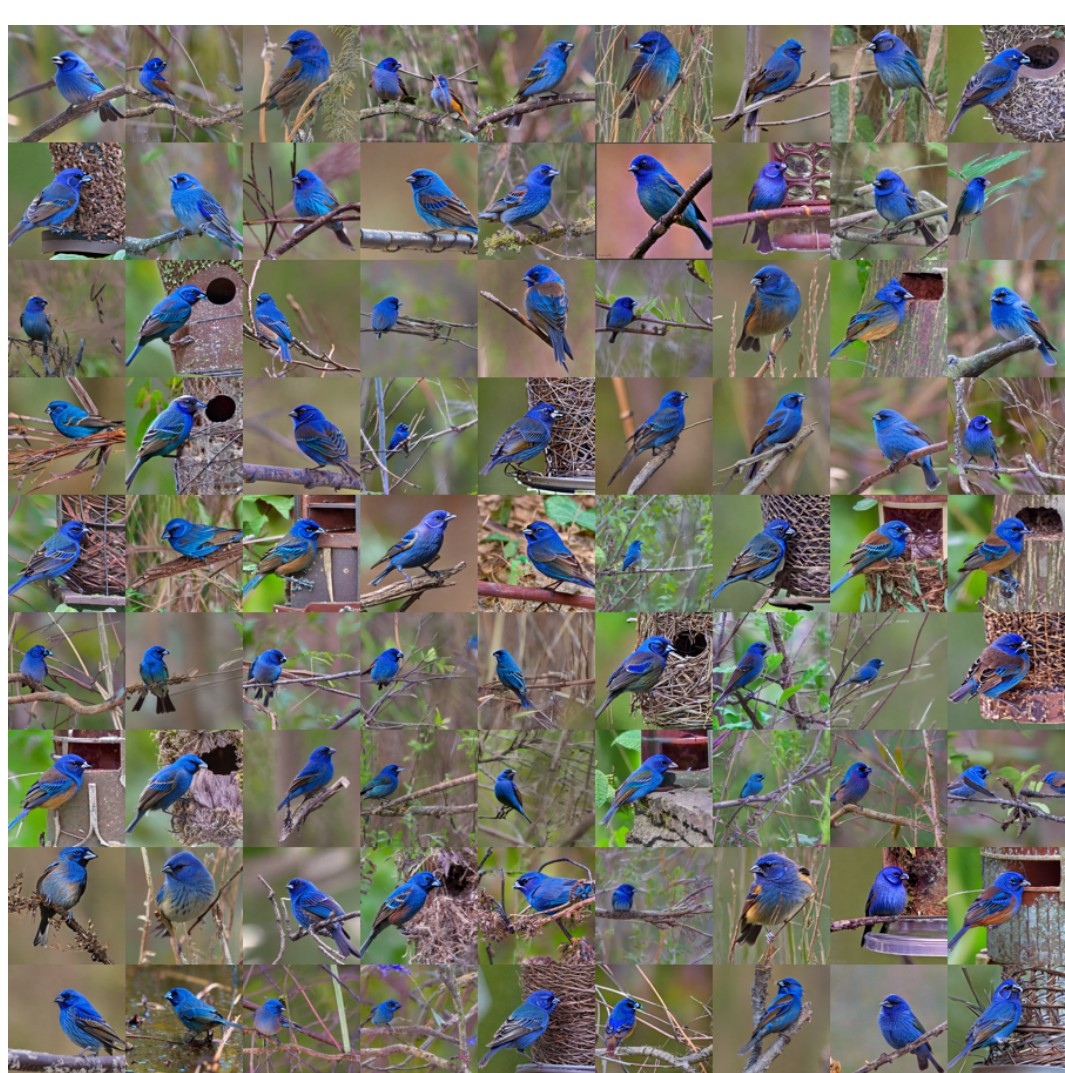

Figure 14: Synthetic data generated for the Indigo Bunting class in the ImageNet dataset using the DPSDA2.

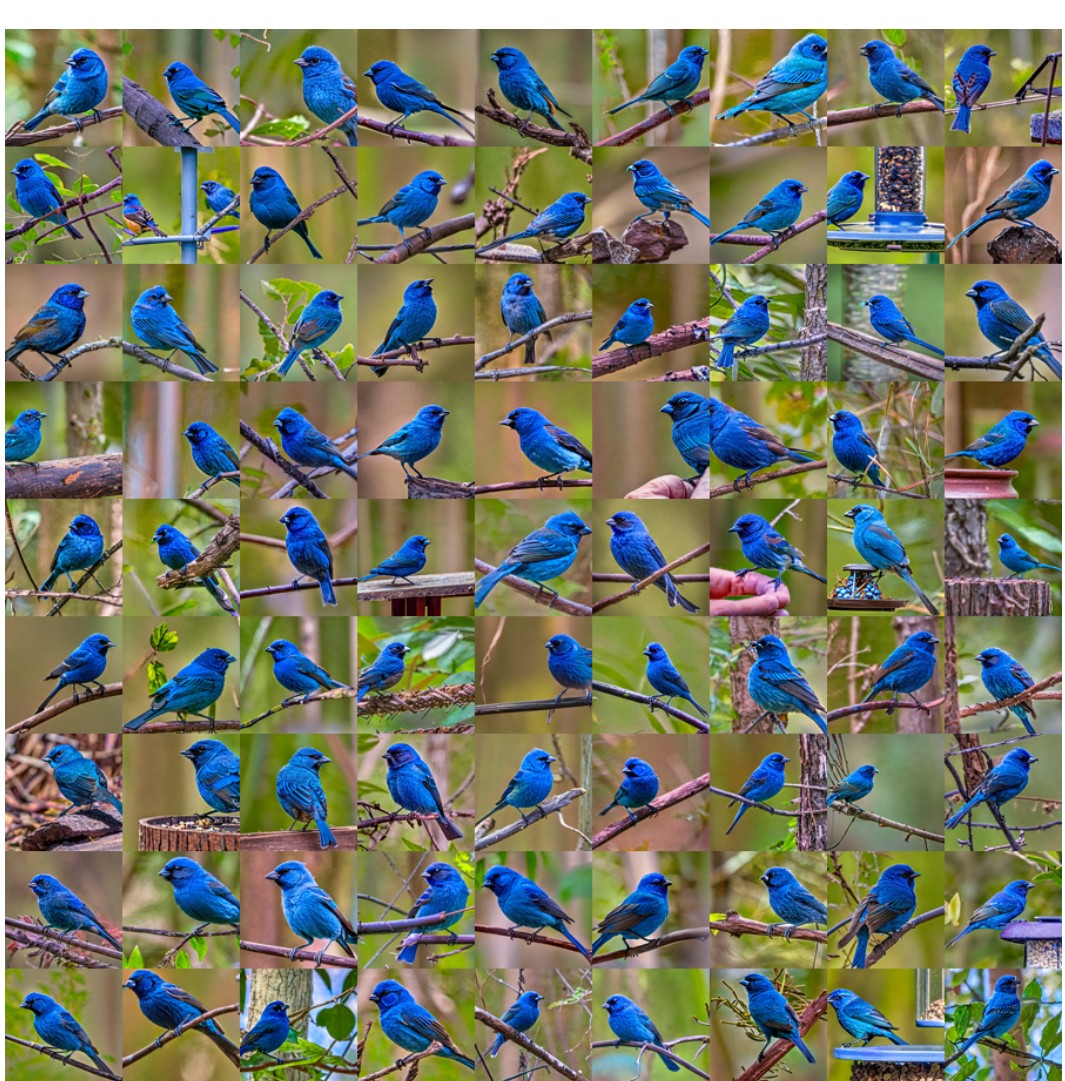

Figure 15: Synthetic data generated for the Indigo Bunting class in the ImageNet dataset using the DPSDivA.

