# OpenReview forum: "Differentially Private Synthetic Data Generation with Diversity via APIs"
_ICLR.cc/2026/Conference — Submitted to ICLR 2026_

### Official Review · Reviewer_U31U · 2025-10-15

**Soundness:** 2
**Presentation:** 2
**Contribution:** 3
**Rating:** 2
**Confidence:** 4

**Summary:**

This paper identifies that previous DP data synthesis method PE tends to repeatedly focus on a limited subset of samples, leading to a significant reduction in the diversity of the generated synthetic dataset. To solve this challenge, Div-PE is proposed by keeping an additional sample for each class. Privacy analysis and converge analysis is all considered. Dataset of multiple modality is included.

**Strengths:**

1.	This paper observes a very important drawback of PE, i.e. the decrease of synthetic data diversity with the use of VARIATE_API on the top-voted synthetic sample within each class. Another sample “non-winning” sample that is not selected by PE is selected and preserved to the next generation round to increase diversity.
2.	Variation prompts are carefully designed to further increase the diversity.
3.	Experiment on 3 modality is included, including image, text and tabular.

**Weaknesses:**

1.	Lack of baseline for tabular data. In Table 1, no baseline method is included for tabular datasets. As GreaT is applied to serialize table rows into natural language, Aug-PE is a direct baseline that can be compared. I would like to see the comparison.
2.	Lack of hyper-parameter selection study. As this is a paper considering differential privacy, one very important hyper-parameter is the differentially privacy budget $\epsilon$. The robustness of the proposed Div-PE under different $\epsilon$ should be studied but is currently missing in the main paper (please move it into the main paper), and lacks the comparison with baselines.
3.	Lack of in-depth study of the proposed Div-PE. Many hyper-parameters are selected without given a logic, i.e. it’s impact on the final performance is not studied. For example, $T, N_{can}$.

I think the overall idea is good and the proposed solution is interesting, but given the lack of experiments and mistakes (see Questions) contained in the paper, I cannot give a positive score for the current version which is not ready for being accepted as a top-tier conference paper. I will increase my score if the revison is good.

**Questions:**

1.	If I understand correctly, in Algorithm 1, each $S_t$ contains $2N_{syn}$ samples as 2 step voting each contribute $N_{syn}$ samples. Therefore, within each iteration, should the number of total candidates for step-1 voting be $2\times N_{syn}\times N_{can}$ but not $N_{syn}\times N_{can}$?
2.	The paper mentioned that, the second voting uses the selected synthetic data instead of private data to avoid privacy budget increasement. Is there any experimental comparison results on this?
3.	I am curious, do you have any more detailed explanation on why removing adaptive variation (in Table 2) results in a sharp increase in FID and a dramatic decrease in Recall? The current analysis is too brief for me to understand.
4.	See weaknesses.

---

> ### Author Response · Authors · 2025-12-01
> **1. Clarification on the Number of Candidates in Algorithm 1**
>
> Thank you for your careful reading of Algorithm 1.
>
> The intention of the algorithm is *not* that each voting stage independently contributes a full set of $k$ new candidates—leading to a total of $2k$—but rather that both stages operate under a **fixed survivor budget** $|S^{(t)}| = k$ for each generation.
>
> Concretely, at every iteration we maintain a fixed number of survivors $k$.
> The first voting stage selects a subset of the next-generation candidates under this budget, and the second stage fills the *remaining slots* using diverse non-winning candidates.
> In other words, the two stages **partition** the same total survivor budget $k$ rather than producing $k$ survivors each.
>
> We understand that the current pseudocode notation could be misread as if each stage were generating an additional $k$ candidates.
> We will revise the algorithm description to make this fixed-budget behavior clearer and avoid such ambiguity.

---

> ### Author Response · Authors · 2025-12-01
> **2. Experimental Comparison of Second Voting with Private vs. Synthetic Data**
>
> Thank you for the question. In this work, we did not include an explicit experimental comparison between a variant that uses private data again in the second voting stage and our design that relies only on synthetic data. Our choice was driven by the privacy analysis: limiting each iteration to a single private-data-based vote simplifies accounting and avoids increasing the per-iteration privacy budget. We agree that evaluating both variants side by side especially in terms of utility and diversity under different privacy budgets would be valuable, and we will consider adding such experiments in a future revision or extended version of the work.

---

> ### Author Response · Authors · 2025-12-01
> **3. Clarification on the effect of removing adaptive variation**
>
> In Table 2, the sharp increase in FID and the large drop in Recall are not caused by removing adaptive variation (av). The numbers show this clearly:
> - auto → auto+av: FID 48.448 → 48.475 (almost unchanged), Recall 0.301 → 0.418 (slightly higher)
> - auto → auto+demo: FID 48.448 → 136.818, Recall 0.301 → 0.083
>
> So the dramatic changes mainly come from adding demonstration (demo), not from av.
> Demonstrations directly guide the variation process, leading to faster convergence but reduced exploration, which lowers diversity and thus harms Recall and FID much more than av does. AV modulates local variation strength, but it does not change the underlying sample-to-sample coupling that demo introduces. This is why the differences between auto and auto+av are modest, whereas the presence of demo causes both the FID spike (due to reduced coverage/diversity) and the Recall drop.

---

> ### Author Response · Authors · 2025-12-01
> **Additional Tabular Baseline**
>
> Regarding the concern about missing tabular baselines, we agree this is an important gap and have updated our evaluation accordingly.
> To address this, we now include results with GReaT, a standard non-DP text-based tabular generator that uses the same row-serialization scheme as our method. This makes it a natural non-DP reference point for Div-PE in the tabular setting. While PE and Aug-PE could in principle be applied to tabular data using the same serialization, in this work we chose to focus on establishing, to our knowledge, the first DP evolutionary framework for tabular data. Extending PE and Aug-PE to this modality would further strengthen the comparison, and we plan to include these variants in a follow-up version.
>
> **Results (Adult / Body-Performance datasets)**
> | Algorithm | Dataset          | Accuracy | Wasserstein ↓ | Recall ↑ | Precision ↑ | Density ↑ | Coverage ↑ |
> |-----------|------------------|----------|---------------|----------|--------------|------------|-------------|
> | ours      | adult            | 0.918    | 0.0167        | 0.7187   | 0.7207       | 0.3397     | 0.6287      |
> | ours      | body-performance | 0.772    | 0.0230        | 0.4363   | 0.4603       | 0.1263     | 0.3937      |
> | GReaT     | adult            | 0.960    | 0.0078        | 0.700    | 0.830        | 0.520      | 0.460       |
> | GReaT     | body-performance | 0.850    | 0.012         | 0.430    | 0.640        | 0.250      | 0.320       |
>
> **Interpretation**
>
> As expected for a non-DP method, GReaT delivers strong fidelity and downstream utility. Importantly, even under strict privacy constraints, Div-PE attains higher coverage on both datasets, indicating that its diversity-preserving mechanism transfers effectively to the tabular domain and can recover a broader portion of the real data distribution than the non-DP baseline, despite the added DP noise.

---

### Official Review · Reviewer_o6x3 · 2025-10-19

**Soundness:** 3
**Presentation:** 3
**Contribution:** 3
**Rating:** 2
**Confidence:** 5

**Summary:**

This paper proposes Div-PE, a privacy-preserving framework for generating synthetic data using black-box foundation model APIs (e.g., Stable Diffusion for images, Llama-2 for text/tabular). It builds on PE, which uses evolutionary algorithms to guide synthetic data toward private distributions without fine-tuning or exposing raw data. However, PE suffers from diversity collapse, where generations converge to variations of a few high-fitness samples, reducing utility in downstream tasks. Div-PE addresses this via a two-stage voting mechanism that promotes balanced selection, ensuring broader ancestral lineages survive while maintaining DP guarantees.

**Strengths:**

- The diversity of synthetic images is important in the DP image synthesis area.

- The paper is easy to follow and well-written.

**Weaknesses:**

- Enhancing prompts to improve generative diversity is still fundamentally constrained by the model's inherent capabilities. This approach does not fundamentally overcome the limitations of the model's generative power. If the model lacks the ability to produce images similar to sensitive ones, such methods will not perform well. The author is encouraged to provide a more in-depth discussion and investigate how Div-PE performs in scenarios where the model's generative capacity is limited.

- The benefits and motivation behind the two-stage generation approach are not clearly articulated. Why does it work? In fact, the quality of synthetic images is generally low compared to real sensitive images, and using synthetic images for voting inherently introduces a lot of noise, which can negatively impact the voting results.

- The method relies on LLMs to generate a diverse prompt set. However, if the public information ($I_pub$ is insufficient or the LLM exhibits bias (e.g., cultural bias), the initial synthetic dataset ($S_0$) may lack diversity, potentially hindering convergence in subsequent iterations. The paper does not provide robustness evaluations, such as performance under noisy public information. Furthermore, both the adaptive mutation (Eq. 3) and demonstration-based mutation (Eq. 2) depend on the voting score ($V^{(1)}$), which may be unstable under noise ($\sigma$), leading to misleading guidance—where superior samples incorrectly influence inferior ones. The paper does not demonstrate the stability of these mechanisms under high noise (i.e., low $\epsilon$) conditions.

- The first stage assumes that each private sample contributes only one vote (i.e., sensitivity = 1). However, in multi-class settings, cross-class interference may increase the actual sensitivity, which is not addressed in the paper.

- Although the method draws inspiration from “natural ecology” to prevent monopolization, it does not model realistic evolutionary dynamics—such as mutation rate decay—which may lead to collapse over long-term iterations.

- This paper does not discuss or compare the approach of using public data for pretraining combined with DP-SGD-based differentially private image synthesis. I’m curious how Div-PE performs relative to such methods.

- The authors did not investigate how different APIs affect the quality of generation.

- Although the authors claim that Div-PE improves the diversity of the generated dataset, the results in Table 1 suggest otherwise. In fact, the recall is lower than the baseline, and the best value is incorrectly marked. Recall is a metric where higher values indicate better performance.

- Moreover, it is unclear how Camelyon17 achieves an accuracy close to 86.1% despite having a very high FID. The same concern applies to ImageNet. These experimental results are not convincing.

**Questions:**

Please refer to the weakness.

---

> ### Author Response · Authors · 2025-12-01
> **Acknowledgment and Response to Key Weaknesses**
>
> Thank you for the detailed and candid review. We have carefully read all of the weaknesses you pointed out. Many of them raise meaningful considerations which we will keep in mind and address more thoroughly in a future revision or extended version of the work. Below, we briefly comment on several of the major points.
>
> # 1. On the limitation that prompt enhancement cannot overcome the model’s inherent generative capability
>
> We fully agree that prompt-based techniques cannot fundamentally exceed the generative ceiling of the underlying foundation model. In our framework, Auto-Prompt is not the core mechanism but an optional enhancement. The primary contribution of Div-PE lies in the BISTAGE procedure.
>
> Indeed, even without Auto-Prompt, BISTAGE alone yields substantially higher utility and diversity than PE (e.g., accuracy improving from ~0.41 to ~0.78, along with corresponding increases in diversity metrics). Auto-Prompt affects variation quality but is not necessary for the gains observed in Div-PE. We will clarify this distinction more explicitly in the paper.
>
> # 2. Motivation and effects of the two-stage selection
>
> Your concern about noise in synthetic samples is well taken. The goal of the two-stage procedure is not to rely on the absolute quality of individual synthetic samples but to rebalance ancestry probabilities so that minority lineages are not prematurely eliminated. This smoothing effect empirically reduces degeneracy in longer iterative runs. We acknowledge that the conceptual motivation could be articulated more clearly and will refine the exposition accordingly.
>
> # 3. Sensitivity to noisy public information and potential instability of the voting score
>
> We agree that LLM bias and noisy auxiliary information can affect Auto-Prompt quality. As mentioned above, the core performance gains of Div-PE do not depend on Auto-Prompt. Although we did not observe instability under lower ε values across repeated runs, we agree that a more formal robustness analysis would strengthen our claims. We will consider this direction in future work.
>
> 4. On the sensitivity assumption in multi-class settings
>
> Although the private dataset may be multi-class, Div-PE (like PE) performs the evolutionary procedure independently per class. Therefore, each class-specific run maintains the same sensitivity bound as the single-class setting, and cross-class interference does not occur within the mechanism. We will clarify this implementation detail to prevent misunderstanding.
>
> # 5. Ecological analogy and mutation dynamics
>
> The ecological analogy was intended as a high-level intuitive motivation, not as an attempt to model real biological evolutionary dynamics. That said, incorporating dynamics such as mutation-rate decay or resource-competition effects could enrich the framework and may yield deeper insights into long-horizon behavior. We consider this an interesting direction and plan to explore such extensions in future work.
>
> # 6. Comparison with DP-SGD–based private generation
>
> We have indeed considered DP-CTGAN and AIM earlier in our exploratory phase, and conducted preliminary experiments on small toy tabular datasets. However, these methods require white-box access to model parameters and directly learn from the full private dataset via DP-SGD or marginal estimation. In contrast, our work focuses on black-box foundation-model–based synthetic data generation, so we intentionally limited our main comparisons to PE-style approaches operating under the same black-box constraint.
>
> That said, we agree that positioning Div-PE relative to tabular DP-SDG methods such as DP-CTGAN or AIM could provide valuable broader context. We will take this into account in a future revision and consider including such comparisons in an extended version of the work.
>
> # 7. Effect of using different APIs
>
> We agree that evaluating additional APIs could provide insight into generality. Due to cost and space constraints, we limited the scope for this submission, but this is a natural direction for future experimentation.
>
> # 8. Interpretation of Table 1 and recall values
>
> Recall increases when the generated set covers more real-data modes. In Div-PE, diversity increases may shift the precision–recall balance, causing recall to decrease in some settings. We will correct the bolding issue and clarify this interpretation.
>
> # 9. On high FID but high downstream accuracy
>
> FID measures distribution-level similarity, whereas downstream accuracy can remain high when the generated data preserve discriminative structures—even if low-level statistics differ. For datasets with strong label–feature correlations (e.g., Camelyon17), this phenomenon is common. We agree that this deserves clearer discussion and will elaborate in a future revision.
>
> Once again, thank you for the thorough review. Your comments highlight several important considerations, and we will incorporate these insights into future refinements of the work.

---

### Official Review · Reviewer_yTep · 2025-10-28

**Soundness:** 3
**Presentation:** 3
**Contribution:** 3
**Rating:** 6
**Confidence:** 3

**Summary:**

The paper introduces Div-PE, a framework for DP synthetic data generation using Private Evolution (PE). Div-PE introduces a two-stage voting mechanism that aims to mitigate the diversity collapse observed in prior PE methods where dominant samples monopolize the final synthetic dataset reducing the overall diversity. This sampling procedure relies on the post-processing property of DP and has no additional privacy cost. Empricially, the authors show Div-PE strengthens both utility and diversity across various modalities including image, text and tabular benchmarks.

**Strengths:**

- The paper is well-presented and clearly written.
- The paper highlights and solves a key limitation of PE regarding diverse data generation and proposes a well-motivated approach to fix this problem.
- The empirical results cover three different modalities (text, image and tabular) across multiple benchmark datasets and show the proposed method consistently improves utility and diversity over standard PE.

**Weaknesses:**

- Comparisons compare only against other PE baselines and existing DP synthetic data baselines are missing (particularly for the tabular setting), making it difficult to contextualize results against the broader DP synthetic data literature.
- While Div-PE claims similar overhead to standard PE, there is no direct measurement of overhead, particularly for the two-stage sampling procedure or the Auto-Prompt generation.
- The role of Auto-Prompt appears dominant in improving diversity, as suggested by Table 2. The contribution of the new two-stage voting seems unclear.

**Questions:**

1. The ablation in Table 2 suggests Auto-Prompt contributes most to diversity improvement. Is this ablation with BISTAGE for every option? I would have preferred to have seen a more detailed ablation across multiple datasets to really understand the contribution of each component. How does PE+Auto compare with just BISTAGE?
2. Related, how does auto-prompt work for tabular data generation? How exactly do you modify the GReaT encoding in this setting?
3. Could the authors provide more detailed runtime results? Specifically, what is the actual overhead of BISTAGE+auto?
4. How does the method perform across different DP budgets? Are diversity gains consistent under stronger privacy?
5. What accounts for the discrepancy between Table 2 and Figure 6 in coverage scores? I can't seem to find what the exact experimental setup (dataset etc.) was used for Figure 6? The Figure 6 ablation seems to imply that auto-prompt gets you most of the way in terms of best FID/coverage trade-off?
6. For the tabular setting, have the authors thought about comparing against existing tabular DP-SDG methods such as DP-CTGAN or SOTA marginal-based methods like AIM?

---

> ### Author Response · Authors · 2025-12-01
> **1. Clarification on Ablation Settings and Component Contributions**
>
> Thank you for the thoughtful questions. We address each point below.
>
> (1) Whether all ablations in Table 2 were conducted under the BISTAGE setting
> Yes — all configurations in Table 2 were evaluated with BISTAGE enabled.
>  The ablations were designed to isolate the effect of the variation-related components (Auto-Prompt, adaptive variation, and demonstration-based variation) on top of the same two-stage selection mechanism.
>
> (2) On providing more detailed ablations across multiple datasets
> We agree that a broader, multi-dataset ablation would provide a more complete understanding of the contribution of each component.
> We appreciate the suggestion and will consider adding more extensive ablations across datasets in a future revision or extended version of the work.
>
> (3) How PE+Auto compares with BISTAGE alone
> To further clarify the role of BISTAGE itself, we conducted an additional supplementary experiment (performed after receiving the review), evaluating BISTAGE alone—without Auto-Prompt, adaptive variation, or demonstration-based variation—on ImageNet.
> Even in this minimal configuration, BISTAGE shows substantial gains over PE- and Aug-PE–style baselines:
>
> - Accuracy: 0.786
> - FID: 54.372
> - Precision: 0.894
> - Recall: 0.142
> - Density: 1.109
> - Coverage: 0.553
>
>
> While these scores are naturally lower than those of the full Div-PE (which includes Auto-Prompt), they still outperform PE and Aug-PE across utility and diversity metrics.
>  This supports our claim that BISTAGE—not Auto-Prompt—is the main driver of the diversity improvements, with Auto-Prompt serving as an optional enhancement.
> Thank you again for the constructive and insightful questions.

---

> ### Author Response · Authors · 2025-12-01
> **2. How Auto-Prompt Works for Tabular Data — Integration with GReaT Encoding**
>
> Our method keeps the original GReaT linearization of a table row completely unchanged.
> Each record is still represented as a flat natural-language sequence:
>
> ```age is 25, workclass is Private, fnlwgt is 226802, …```
>
> Auto-prompt does not modify this schema.Instead, it injects a record-specific system prompt before the GReaT prompt. This system prefix summarizes a small set of key attributes so that each record is generated under a slightly different semantic guidance.
> Example of the actual inference-time input:
>
> ```
> <SYSTEM> Individual description: a young male working full-time with no capital gains/losses.</SYSTEM>
> age is 25, workclass is Private, fnlwgt is <mask>, education is 11th..
> ```
>
>
> Thus:
> - Schema stays identical to GReaT (“key is value”).
> - Only the prefix changes, enabling per-record diversity.
> - Decoding remains trivial, as the structure of the output never changes.

---

> ### Author Response · Authors · 2025-12-01
> **3. Clarification on Runtime and Overhead Reporting**
>
> Thank you for raising this point.
> We agree that reporting more detailed runtime measurements—especially for BISTAGE+Auto would help provide a clearer picture of the computational overhead.
>
> For this submission, our focus was primarily on the algorithmic behavior and diversity/utility outcomes, so we did not include extended runtime profiling. We will consider providing more comprehensive runtime analysis in a future revision or extended version of the work, where space and scope allow for a fuller discussion.
>
> Thank you again for the helpful suggestion.

---

> ### Author Response · Authors · 2025-12-01
> **4. Impact of Different DP Budgets on Diversity**
>
> The diversity improvements primarily stem from the BISTAGE selection mechanism itself, rather than from the privacy level. Adjusting the DP budget mainly affects the noise added to the first-stage voting scores, not the structural behavior of the two-stage process. Therefore, we expect the overall diversity benefits to remain largely consistent even under stronger privacy (smaller ε).
>
> That said, higher noise can introduce degradation in sample quality or fitness estimation, which may  affect downstream performance. We will include a more explicit discussion of this effect in a future revision.

---

> ### Author Response · Authors · 2025-12-01
> **5. Clarification on the Coverage Discrepancy Between Table 2 and Figure 6**
>
> The discrepancy mainly comes from the experimental setup: Table 2 averages results over five ImageNet classes, while Figure 6 is computed on only two classes. Since metrics like Coverage and KID are highly sensitive to class composition, using different subsets naturally produces different numeric ranges.
>
> Figure 6 can therefore make auto-prompt look sufficient for those specific classes, but the five-class average in Table 2 shows that adding demo/av yields more stable improvements overall. The difference reflects class-sensitivity, not inconsistency.

---

> ### Author Response · Authors · 2025-12-01
> **6. Future Consideration of DP-CTGAN/AIM Comparisons in Tabular Settings**
>
> Thank you for the suggestion. We focused our experiments on PE-style baselines that operate under the same overall assumptions as Div-PE, so tabular DP-SDG methods such as DP-CTGAN or AIM were not included in this version. We agree that such comparisons could provide additional context, and we will keep this in mind for a future revision or an extended version of the work.

---

### Official Review · Reviewer_5dWP · 2025-10-30

**Soundness:** 3
**Presentation:** 1
**Contribution:** 4
**Rating:** 4
**Confidence:** 4

**Summary:**

The paper proposes an improvement to a previous private evolution (PE) method to generate DP synthetic data using a foundation model API, with the aim of improving the diversity of the generated samples. The issue with the existing method is that all of the generated samples become very similar with many iterations of the algorithm, since they all become variations of a single example. The paper's solution, called Div-PE, is to ensure that variations of all initial examples are kept throughout the process. This is done by adding an additional step on top of the selection step of PE where poorly-performing examples are selected based on their similarity to well-performing examples. The second step uses the same DP statistics than PE, so it does not bring extra privacy cost. Div-PE is compared against PE and another variant of it on 7 datasets of image, text and tabular data. Div-PE outperforms the others in almost all settings.

**Strengths:**

The paper identifies a clear weakness in PE and provides a novel solution, which is demonstrated to solve the problem. Synthetic data generation with DP is an important problem, and algorithms effectively using API-only foundation models are especially useful due to the prevalence of these foundation models. Previous works in this area have not generated tabular data, which is a welcome addition in this paper.

**Weaknesses:**

The paper is missing many important explanations that are needed to fully understand the results. There are also many minor issues that together significantly reduce the clarity of the paper. In particular, Section 3 should explain SEED_API, VARIATION_API, the distance function $d$ and how different degrees of variation can be obtained from VARIATION_API in practice. Also, many important experimental details are missing. These are critical for reproducing the experiments and fully understanding their significance:
- Foundation model for tabular data is not specified.
- Not clear whether precision and recall in Table 1 are classification metrics or synthetic data evaluation metrics.
- It is not clear how SEED_API, VARIATION_API and the distance function are implemented in the experiments, or how different degrees of variation are obtained from VARIATION_API.
- Results do not have uncertainty estimates.

The paper is also missing comparisons with two recently published baselines improving the original PE: Tan et al. (2025) and Zhang et al. (2025). Another baseline that should be included is a conventional DP tabular data generator such as AIM (McKenna et al. 2022) for the tabular datasets.

Minor points:
- Figure 1: text is too small, and panel (a) is smaller than the others for some reason.
- Not clear why only one lineage surviving is important based on Figure 1. The synthetic data in panel (b) seems to have slightly better diversity than in panel (c). Figure 2 does a much better job of communicating the issue.
- Lines 147-148: "differ by at most one individual" is ambiguous. It could mean that one individual is added or removed, or that one individual is changed.
- Algorithm 2, line 14: should $u$ be used instead of $j$? Also, $S\_{syn}$ is not defined.
- Line 256: I don't understand what "vote within their own groups" means. Based on Algorithm 2, it looks like all the selected samples just vote together.
- Algorithm 1, line 13: the arguments are in a different order than the parameters of Algorithm 2. Also, the distance function is missing.
- Equations (2) and (3) are not reflected in Algorithms 1 and 2.
- Table 1: not all best values are bolded, for example recall on ImageNet.
- The proposed algorithm is called "DPSDivA" in Appendix C.2.

References:
- R. McKenna, B. Mullins, D. Sheldon, G. Miklau (2022) "AIM: an Adaptive and Iterative Mechanism for Differentially Private Synthetic Data" Proceedings of the VLDB Endowment
- B. Tan, Z. Xu, E. P. Xing, Z. Hu, S. Wu. (2025) "Synthesizing Privacy-Preserving Text Data via Finetuning *without* Finetuning Billion-Scale LLMs" ICML
- J. Zhang, Y. Liu, J. Fu, Y. Hua, T. Zou, J. Cao, Q. Yang (2025) "PCEvolve: Private Contrastive Evolution for Synthetic Dataset Generation via Few-Shot Private Data and Generative APIs" ICML

**Questions:**

- Why does Private-PE degenerate into generating variants of a single example? Intuitively, candidates from more than one example should always get selected, since variants of a single example can only be nearest to a subset of training samples, while variants of another example would be nearest to another subset.

---

> ### Author Response · Authors · 2025-12-01
> **Why Private-PE Collapses — Diversity Decline Mechanism**
>
> To make the explanation clearer, we include the following illustrative diagram.
>
> Figure: Why Private-PE Collapses — Diversity Decline Mechanism
> (Source: https://i.postimg.cc/SNGqyh3k/How-Private-PE-Collapses-to-a-Single-Ancestral-Lineage.png)
>
> The diagram illustrates how repeated cycles of variation and voting gradually concentrate selection on a single high-fitness ancestor in Private-PE. As a result, other lineages gradually lose probability mass and are eliminated, and the synthetic population collapses into variants of that single ancestor — explaining the sharp decline in diversity observed in PE.
>
> We have carefully read all of the weaknesses you pointed out, and we sincerely appreciate the level of detail in your review. Many of your observations will help us clarify the exposition and improve the overall presentation. We will take them into account and refine the manuscript accordingly in a future revision. Thank you again for your thoughtful and constructive comments.

---

### Meta-Review · Area_Chair_DVie · 2025-12-09

**Summary:**

The paper proposes an extension of the Private Evolution (PE) framework in Lin et al., 2023 to generate DP synthetic data using a foundation-model API, with the goal of improving sample diversity. The paper introduces Div-PE to ensure that variations of all initial examples are kept throughout the process. However, reviewers have identified several weaknesses. The authors partially answered some of the questions from reviewers but still keep many other questions not answered. For example, Reviewer 5dWP provided many thoughtful comments (e.g., missing baselines and technical details) but the authors only acknowledge these comments without providing any responses.

**Reviewer Concerns:**

First, they found the experimental results to be relatively weak: the paper focuses mainly on image data, lacks experiments on tabular data, and includes no baselines beyond the original PE method. During the rebuttal period, the authors wrote “We focused our experiments on PE-style baselines that operate under the same overall assumptions as Div-PE, so tabular DP-SDG methods such as DP-CTGAN or AIM were not included in this version”. However, I believe it is important to compare this line of work to highlight the significance of PE-based methods compared with marginal/workload based methods in DP synthetic tabular data literature.

Second, the writing could be improved. Many technical details are missing, making the method difficult to follow. Finally, the paper would benefit from a more comprehensive discussion of its limitations. For example, as reviewers mentioned, the proposed approach relies heavily on the capabilities of the chosen foundation model. If the model is unable to generate samples that resemble the sensitive data, the effectiveness of the method may be significantly limited.

**Reviewer Scores:**

As I wrote above, the authors partially answered some of the questions from reviewers but still keep many other questions not answered. For example, Reviewer 5dWP provided many thoughtful comments (e.g., missing baselines and technical details) but the authors only acknowledge these comments without providing any responses. Hence, I would guess the reviewers wouldn't change their scores if they had been able to engage in the rebuttal period.

---

### Decision · Program_Chairs · 2026-01-26

Reject